# End-to-End Conformal Calibration for Optimization Under Uncertainty

**Christopher Yeh**[*]                                                          *cyeh@caltech.edu*
**Nicolas Christianson**[*]                                                    *nchristi@caltech.edu*
**Alan Wu**                                                                     *ywu9@caltech.edu*
**Adam Wierman**                                                                *adamw@caltech.edu*
**Yisong Yue**                                                                  *yyue@caltech.edu*
*Department of Computing and Mathematical Sciences*
*California Institute of Technology*

**Reviewed on OpenReview:** *https://openreview.net/forum?id=yM8qkTOf9H*

## Abstract

Machine learning can significantly improve performance for decision-making under uncertainty across a wide range of domains. However, ensuring robustness guarantees requires well-calibrated uncertainty estimates, which can be difficult to achieve with neural networks. Moreover, in high-dimensional settings, there may be many valid uncertainty estimates, each with its own performance profile—i.e., not all uncertainty is equally valuable for downstream decision-making. To address this problem, this paper develops an end-to-end framework to *learn* uncertainty sets for conditional robust optimization in a way that is informed by the downstream decision-making loss, with robustness and calibration guarantees provided by conformal prediction. In addition, we propose to represent general convex uncertainty sets with partially input-convex neural networks, which are learned as part of our framework. Our approach consistently improves upon two-stage estimate-then-optimize baselines on concrete applications in energy storage arbitrage and portfolio optimization.

## 1 Introduction

Well-calibrated estimates of forecast uncertainty are vital for risk-aware decision-making in many real-world systems. For instance, grid-scale battery operators forecast electricity prices to schedule battery charging/discharging to maximize profit, while accounting for forecast uncertainty to mitigate financial or operational risk. Similarly, financial investors use forecasts of asset returns with uncertainty estimates to maximize portfolio returns while minimizing downside risk.

Traditional approaches to decision-making under uncertainty often follow an "estimate-then-optimize" (ETO) paradigm (Chenreddy et al., 2022), in which the uncertainty estimation and decision-making stages are decoupled. In the "estimation" stage, a predictive model is trained to forecast a target quantity along with an uncertainty set. In the "optimization" stage, this uncertainty estimate informs a downstream decision. Crucially, the cost or performance of the downstream decision is typically not fed back into the training of the predictive model.

A recent line of work (Chenreddy et al., 2022; Patel et al., 2024; Wang et al., 2024; Chenreddy & Delage, 2024) has made steps toward bridging the gap between uncertainty quantification and robust optimization-driven decision-making, where optimization problems take a forecast uncertainty set as a parameter, as is common in energy systems (Yan et al., 2020; Parvar & Nazaripouya, 2022) and financial applications (Gregory et al., 2011; Quaranta & Zaffaroni, 2008). However, existing approaches are suboptimal for several reasons:

---

[*]These authors contributed equally.

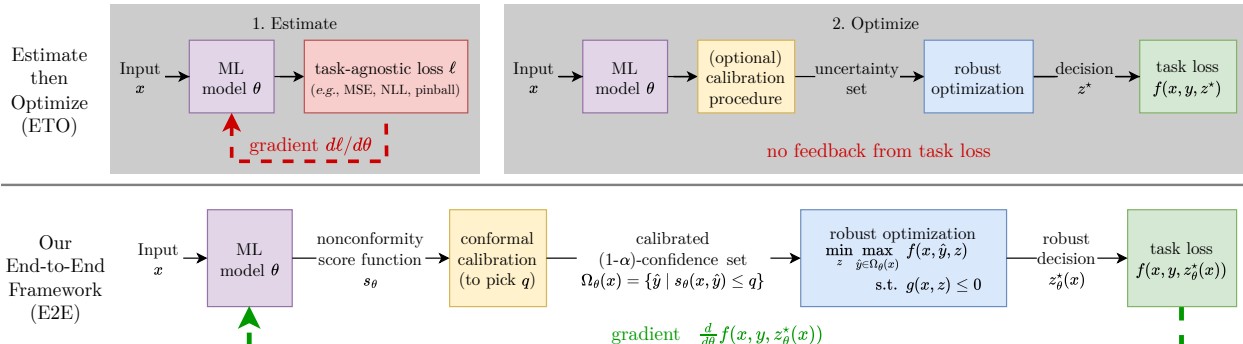

Figure 1: Whereas prior "estimate-then-optimize" (ETO, top) approaches separate the model training from the optimization (decision-making) procedure, we propose a framework for end-to-end (E2E, bottom) conformal calibration for optimization under uncertainty that directly trains the machine learning model using gradients from the task loss.

1. **Lack of decision-aware training:** The predictive model is typically not trained with feedback from the downstream objective. Because the downstream objective is often asymmetric with respect to the forecasting model's errors, prediction models trained with decision-agnostic losses may achieve high predictive accuracy but still perform poorly on the decision-making objective.

2. **Restricted uncertainty set parameterizations:** To preserve tractability of the robust optimization problem, uncertainty sets are often constrained to simple parametric forms (e.g., boxes or ellipsoids), limiting the expressivity of uncertainty estimates.

3. **Calibration challenges:** Because neural network models are often poor at estimating their own uncertainty, the forecasts may not be well-calibrated. Recent approaches such as isotonic regression (Kuleshov et al., 2018) and conformal prediction (Shafer & Vovk, 2008) have made progress in providing *calibrated* uncertainty estimates from deep learning models, but such methods are typically applied post-hoc to trained models and are therefore difficult to incorporate into an end-to-end training procedure.

As such, there is as of yet *no* comprehensive methodology for training calibrated uncertainty-aware deep learning models end-to-end with downstream decision-making objectives. In this work, we provide the first such methodology. We make three specific contributions corresponding to the three issues identified above:

1. **We develop a framework for training prediction models end-to-end with downstream decision-making objectives and conformal-calibrated uncertainty sets in the context of the *conditional robust optimization* problem.** This framework is illustrated in Figure 1 (bottom). By including differentiable conformal calibration in our model during training, we close the loop and ensure that feedback from the uncertainty's impact on the downstream objective is accounted for in the training process, since not all model errors nor uncertainty estimates will result in the same downstream cost. This end-to-end training enables the model to focus its learning capacity on minimizing error and uncertainty on outputs with the largest decision-making cost, with more leeway for outputs that have lower costs.

2. **We propose using partially input-convex neural networks (PICNNs) as the nonconformity score function for conformal prediction, enabling the approximate parametrization of arbitrary compact, convex uncertainty sets in the conditional robust optimization problem.** To the best of our knowledge, no existing works use PICNNs to parametrize such arbitrary convex uncertainty sets. Due to the universal convex function approximation property these networks enjoy (Chen et al., 2019), this approach enables training much more general representations of uncertainty than prior works have considered, which in turn yields substantial improvements on downstream decision-making performance. Importantly, PICNNs are well-matched to our conditional robust optimization problem: we show that the robust problem resulting from this parametrization can be reformulated as a tractable convex optimization problem.

3. **We propose an exact and computationally efficient method to differentiate through the conformal prediction procedure during training.** Unlike prior work (Stutz et al., 2022), our method gives exact gradients, without using approximate ranking and sorting methods.

Finally, we evaluate the performance of our approach on two applications: an energy storage arbitrage task and a portfolio optimization problem. We demonstrate that the combination of end-to-end training with the flexibility of the PICNN-based uncertainty sets consistently improves over ETO baseline methods. The performance benefit of our end-to-end method is apparent even under distribution shift.

## 2 Problem Statement and Background

Our problem is defined formally as follows: suppose that data $(x, y) \in \mathbb{R}^m \times \mathbb{R}^n$ is sampled i.i.d. from an unknown joint distribution $\mathcal{P}$. Upon observing the input $x$ (but not the label $y$), an agent makes a decision $z \in \mathbb{R}^p$. After the decision is made, the true label $y$ is revealed, and the agent incurs a *task loss* $f(x, y, z)$, for some known task loss function $f : \mathbb{R}^m \times \mathbb{R}^n \times \mathbb{R}^p \to \mathbb{R}$. In addition, the agent's decision must satisfy a set of joint constraints $g(x, z) \leq 0$ coupling $x$ and $z$.

As an illustrative example, consider an agent who would like to minimize the costs of charging and discharging a battery over 24 hours in a day. The agent may use weather forecasts and historical observations $x$ to predict future energy prices $y$. Based on the predicted prices, the agent decides on the amount $z$ to charge or discharge the battery. The task loss $f$ is the cost incurred by the agent, and the constraints $g$ include limits on how fast the battery can charge as well as the maximum capacity of the battery. This example is explored in more detail in Section 5.

Because the agent does not observe the label $y$ prior to making its decision, ensuring good performance and constraint satisfaction requires that the agent makes decisions $z$ that are *robust* to the various outcomes of $y$. A common objective is to choose $z$ to robustly minimize the task loss and satisfy the constraints over all realizations of $y$ within a $(1 - \alpha)$-confidence region $\Omega(x) \subset \mathbb{R}^n$ of the true conditional distribution $\mathcal{P}(y \mid x)$, where $\alpha \in (0, 1)$ is a fixed risk level based on operational requirements. In this case, the agent's robust decision can be expressed as the optimal solution to the following **conditional robust optimization (CRO)** problem (Chenreddy et al., 2022):

$$z^\star(x) := \underset{z \in \mathbb{R}^p}{\arg\min} \ \underset{\hat{y} \in \Omega(x)}{\max} \ f(x, \hat{y}, z) \ \text{s.t.} \ g(x, z) \leq 0. \tag{1}$$

After the agent decides $z^\star(x)$, the true label $y$ is revealed, and the agent incurs the task loss $f(x, y, z^\star(x))$. Thus, the agent seeks to minimize expected task loss

$$\mathbb{E}_{(x,y) \sim \mathcal{P}} \left[ f(x, y, z^\star(x)) \right]. \tag{2}$$

While the joint distribution $\mathcal{P}$ is unknown, we assume that we have a dataset $D = \{(x_i, y_i)\}_{i=1}^N$ of i.i.d. samples from $\mathcal{P}$. Then, our objective is to train a machine learning model to learn an approximate $(1 - \alpha)$-confidence set $\Omega(x)$ of possible $y$ values for each input $x$. Formally, our learned $\Omega(x)$ should satisfy the following *marginal coverage* guarantee.

**Definition 2.1** (marginal coverage). An uncertainty set $\Omega(x)$ for the distribution $\mathcal{P}$ provides *marginal coverage at level* $(1 - \alpha)$ if $\mathbb{P}_{(x,y) \sim \mathcal{P}} (y \in \Omega(x)) \geq 1 - \alpha$.

**Comparison to related work.** The problem of constructing data-driven and machine-learned uncertainty sets with probabilistic coverage guarantees for use in robust optimization has been widely explored in prior literature (e.g., Bertsimas & Thiele (2006); Bertsimas et al. (2018); Alexeenko & Bitar (2020); Goerigk & Kurtz (2023)). Chenreddy et al. (2022) first coined the phrase "conditional robust optimization" for the problem (1) and considered learning context-dependent uncertainty sets $\Omega(x)$ in this setting. However, their approach results in a mixed integer optimization that is intractable to solve for large-scale problems. Moreover, they follow the "estimate then optimize" (ETO) paradigm (Elmachtoub et al., 2023). As shown in Figure 1 (top), the ETO paradigm separates the machine learning model training from the decision optimization. The lack of feedback from the downstream task loss during model training in ETO generally

leads to uncertainty sets $\Omega(x)$ which yield suboptimal results. Several other recent papers follow the ETO paradigm using homoskedastic ellipsoidal uncertainty sets (Johnstone & Cox, 2021), heteroskedastic box and ellipsoidal uncertainty sets (Sun et al., 2023), and a "union of balls" parametrization of uncertainty (Patel et al., 2024). In our experiments (Section 5), we demonstrate consistent improvements over the methods of Johnstone & Cox (2021) and Sun et al. (2023).

Closest to our work is an "end-to-end" formulation of the CRO problem posed by Chenreddy & Delage (2024), which aims to learn conditional uncertainty sets $\Omega(x)$ using a weighted combination of the downstream task loss along with a "conditional coverage loss" to promote calibrated uncertainty. However, they focus solely on ellipsoidal uncertainty sets, and their conditional coverage loss does not provably ensure coverage for their learned uncertainty sets. In our experiments, we do not compare against Chenreddy & Delage (2024) because we only consider other methods with a provable coverage guarantee.[1]

A concurrent related work by Wang et al. (2024) approaches the problem of learning *unconditional* uncertainty sets for robust optimization, while achieving robust constraint satisfaction guarantees. While their method also uses an end-to-end task loss, we find their use of the same uncertainty set $\Omega$ for every problem instance (i.e., $\Omega$ is independent of $x$) to be highly restrictive. For example, in the context of our battery control problem, this restriction would disallow the use of weather forecasts and historical price data to estimate uncertainty in future energy prices.[2] They also use restrictive uncertainty set parametrizations such as box, ellipsoidal, and polyhedral uncertainty. Moreover, their guarantee on constraint satisfaction requires an assumption on the convergence of their algorithm to an approximate stationary point; however, this convergence is only guaranteed asymptotically, and not in finitely many samples.

An alternative line of research designs post-hoc conformal prediction sets aimed at minimizing decision costs in robust classification problems (Kiyani et al., 2025; Cortes-Gomez et al., 2024). However, to date, it is unclear how to adapt these methods, which produce discrete uncertainty sets, for regression settings.

In contrast, our work overcomes these limitations: we incorporate differentiable conformal calibration *during training* to ensure that uncertainty is learned end-to-end in a manner that is both calibrated and minimizes task loss. We apply split conformal post-hoc calibration during inference for provable guarantees on coverage. Furthermore, we use partially input-convex neural networks (Amos et al., 2017) to directly parameterize the nonconformity score function in conformal prediction, enabling a general and expressive representation of arbitrary conditional convex uncertainty regions that can vary with $x$ and be used efficiently in robust optimization.

Beyond the above closely related work, this paper builds upon and contributes to several different areas in machine learning and robust optimization; see Appendix B for a comprehensive discussion.

## 3 End-to-End Training of Conformally Calibrated Uncertainty Sets

In this section, we describe our proposed methodological framework for end-to-end task-aware training of predictive models with conformally calibrated uncertainty for the conditional robust optimization problem (1). Our overarching goal is to learn uncertainty sets $\Omega(x)$ which provide $(1 - \alpha)$ coverage for any choice of $\alpha \in (0, 1)$, and which offer the lowest possible task loss (2). To this end, we must consider three primary questions:

1. How should the family of uncertainty sets $\Omega(x)$ be parametrized?
2. How can we guarantee that the uncertainty set $\Omega(x)$ provides coverage at level $1 - \alpha$?
3. How can the uncertainty set $\Omega(x)$ be learned to minimize expected task loss?

Figure 1 (bottom) illustrates the key parts of our framework to answer these questions. First, we use a machine learning model to parametrize a nonconformity score function $s_\theta$, and we define the uncertainty set $\Omega(x)$ to be a $q$-sublevel set of $s_\theta(x, \cdot)$. Second, we use conformal calibration to pick $q$ to enforce marginal

---

[1]As of the time of writing, the approach of Chenreddy & Delage (2024) also suffers from substantial inconsistencies between their code implementation and the equations from their paper. In particular, the conditional coverage loss proposed in their paper is not implementable, as it will (almost surely) yield zero gradients.

[2]After the present paper's submission, the authors of Wang et al. (2024) updated their preprint and expanded their formulation to accommodate conditional uncertainty sets.

coverage. Third, we backpropagate gradients through both the robust optimization and conformal calibration steps to update the machine learning model, thereby enabling end-to-end learning. Sections 3.1 to 3.3 describe each of these parts in detail, and Algorithm 1 shows pseudocode for both training and inference.

For the rest of the paper, we make the following assumptions on the functions $f$ and $g$ to ensure tractability of the resulting optimization problem.

**Assumption 3.1.** We assume the task loss has the form $f(x, y, z) = y^\top F(x, z) + \tilde{f}(x, z)$, where $F(x, z)$ is an affine function of $z$ and $\tilde{f}(x, z)$ is convex in $z$. Furthermore, we assume that $g(x, z)$ is convex in $z$.

Technically, the assumption on $f$ can be relaxed to the more general setting where $f$ is a *conic-representable saddle function* that is convex in $z$ and concave in $y$ (Juditsky & Nemirovski, 2022), which admits an automated dual representation (Schiele et al., 2023). However, we believe the core ideas for our exposition and proofs are much more straightforward with the stronger assumption in Assumption 3.1.

### 3.1 Representations of the uncertainty set

We consider convex uncertainty sets of the form

$$\Omega_\theta(x) = \{\hat{y} \in \mathbb{R}^n \mid s_\theta(x, \hat{y}) \leq q\}, \tag{3}$$

where $s_\theta : \mathbb{R}^m \times \mathbb{R}^n \to \mathbb{R}$ is an arbitrary *nonconformity score function* that is convex in $\hat{y}$, $q$ is a scalar, and $\theta$ collects the parameters of a model that we will seek to learn. Note that this representation loses no generality; *any* family of convex sets $\Omega(x)$ can be represented as such a collection of sublevel sets of a partially input-convex function $s(x, \hat{y})$. This particular representation is chosen due to the ease of calibrating sets of this form via conformal prediction to ensure marginal coverage, as we will describe in Section 3.2.

In choosing a particular score function $s_\theta$, one must balance two considerations: first, the *generality* of the sets $\Omega_\theta(x)$ that $s_\theta$ can represent, and second, the *tractability* of the resulting robust optimization problem (1). We will now show that our representation (3) generalizes commonly-used box and ellipsoidal uncertainty sets, which are known to have tractable robust problems; later, in Section 4, we will propose to approximate more general convex uncertainty sets using partially input-convex neural networks.

**Box uncertainty sets.** A simple uncertainty representation is box uncertainty where $\Omega(x) = [\underline{y}(x), \overline{y}(x)]$ is an $n$-dimensional box whose lower and upper bounds depend on $x$. Let $h_\theta : \mathbb{R}^m \to \mathbb{R}^n \times \mathbb{R}^n$ be a neural network that estimates lower and upper bounds: $h_\theta(x) = (h_\theta^{\text{lo}}(x), h_\theta^{\text{hi}}(x))$. To represent a box uncertainty set in the form (3), we define a nonconformity score function that generalizes scalar conformalized quantile regression (Romano et al., 2019):

$$s_\theta(x, y) = \max(\left\|h_\theta^{\text{lo}}(x) - y\right\|_\infty, \left\|y - h_\theta^{\text{hi}}(x)\right\|_\infty).$$

Then, the uncertainty set (3) becomes

$$\Omega_\theta(x) = \left[h_\theta^{\text{lo}}(x) - q\mathbf{1}, \ h_\theta^{\text{hi}}(x) + q\mathbf{1}\right] =: \left[\underline{y}(x), \ \overline{y}(x)\right].$$

Given a box uncertainty set $\Omega_\theta(x)$, we can take the dual of the inner maximization problem (see Appendix C.1) to transform the robust optimization problem (1) into an equivalent form that is convex, and hence tractable, under Assumption 3.1:

$$
\begin{aligned}
z_\theta^\star(x) = \arg\min_{z \in \mathbb{R}^p} \min_{\nu \in \mathbb{R}^n} \quad & (\overline{y}(x) - \underline{y}(x))^\top \nu + \underline{y}(x)^\top F(x, z) + \tilde{f}(x, z) \\
\text{s.t.} \quad & \nu \geq \mathbf{0}, \ \nu - F(x, z) \geq \mathbf{0}, \ g(x, z) \leq 0.
\end{aligned}
\tag{4}
$$

**Ellipsoidal uncertainty sets.** Another common form of uncertainty set is ellipsoidal uncertainty. Suppose a neural network model $h_\theta : \mathbb{R}^m \to \mathbb{R}^n \times \mathbb{S}_+^n$ predicts mean and covariance parameters $h_\theta(x) = (\mu_\theta(x), \Sigma_\theta(x))$, so that $\hat{\mathcal{P}}(y \mid x; \theta) = \mathcal{N}(y \mid \mu_\theta(x), \Sigma_\theta(x))$ denotes a predicted conditional density, where $\mathcal{N}(\cdot \mid \mu, \Sigma)$ is the

---

**Algorithm 1** End-to-end conformal calibration for robust decisions under uncertainty

---

**function** TRAIN(training data $D = \{(x_i, y_i)\}_{i=1}^N$, uncertainty level $\alpha$, initial model parameters $\theta$)
    **for** mini-batch $B \subset \{1, \ldots, N\}$ **do**
        Randomly split batch: $B = (B_{\text{cal}}, B_{\text{pred}})$
        Compute $q = \text{QUANTILE}(\{s_\theta(x_i, y_i)\}_{i \in B_{\text{cal}}}, \ 1 - \alpha)$
        **for** $i \in B_{\text{pred}}$ **do**
            Solve for robust decision $z_\theta^\star(x_i)$ using (4), (5), or (8)
            Compute gradient of task loss: $d\theta_i = \partial f(x_i, y_i, z_\theta^\star(x_i))/\partial\theta$
        Update $\theta$ using gradients $\sum_{i \in B_{\text{pred}}} d\theta_i$

**function** INFERENCE(model parameters $\theta$, calibration data $D_{\text{cal}} = \{(x_i, y_i)\}_{i=1}^M$, uncertainty level $\alpha$, input $x$)
    Compute $q = \text{QUANTILE}(\{s_\theta(\tilde{x}, \tilde{y})\}_{(\tilde{x},\tilde{y}) \in D_{\text{cal}}}, \ 1 - \alpha)$
    **return** robust decision $z_\theta^\star(x)$ using (4), (5), or (8)

**function** QUANTILE(scores $S = \{s_i\}_{i=1}^M$, level $\beta$)
    $s_{(1)}, \ldots, s_{(M+1)} = \text{SORTASCENDING}(S \cup \{+\infty\})$
    **return** $s_{(\lceil (M+1)\beta \rceil)}$

---

multivariate normal density function. In this case, we define the nonconformity score function based on the squared Mahalanobis distance (Johnstone & Cox, 2021; Sun et al., 2023)

$$s_\theta(x, y) = (y - \mu_\theta(x))^\top (\Sigma_\theta(x))^{-1} (y - \mu_\theta(x)),$$

which yields uncertainty sets (3) that are ellipsoidal:

$$\Omega_\theta(x) = \{\hat{y} \mid (\hat{y} - \mu_\theta(x))^\top (\Sigma_\theta(x))^{-1} (\hat{y} - \mu_\theta(x)) \leq q\}.$$

Let $L_\theta(x)$ denote the unique lower-triangular Cholesky factor of $\Sigma_\theta(x)$ (i.e., $\Sigma_\theta(x) = L_\theta(x)L_\theta(x)^\top$). Taking the dual of the inner maximization problem and invoking strong duality (see Appendix C.2), we transform the robust optimization problem (1) into an equivalent form that is convex, and hence tractable, under Assumption 3.1:

$$
\begin{aligned}
z_\theta^\star(x) = \underset{z \in \mathbb{R}^p}{\arg\min} \quad & \sqrt{q}\|L_\theta(x)^\top F(x, z)\|_2 + \mu_\theta(x)^\top F(x, z) + \tilde{f}(x, z) \\
\text{s.t.} \quad & g(x, z) \leq 0,
\end{aligned}
\tag{5}
$$

### 3.2 Conformal uncertainty set calibration

As long as the uncertainty set $\Omega_\theta(x)$ can be expressed in the form (3), we can use the split conformal prediction procedure at inference time to choose a value $q$ that ensures $\Omega_\theta(x)$ provides marginal coverage (Definition 2.1) at any confidence level $1 - \alpha$. The split conformal procedure assumes access to a calibration dataset $D_{\text{cal}} = \{(x_i, y_i)\}_{i=1}^M$ drawn exchangeably from $\mathcal{P}$. We refer readers to Angelopoulos et al. (2023) for details on this procedure.

**Lemma 3.2** (from Angelopoulos et al. (2023), Appendix D). *Let $D_{cal} = \{(x_i, y_i)\}_{i=1}^M$ be a calibration dataset drawn exchangeably (e.g., i.i.d.) from $\mathcal{P}$, and let $s_i = s_\theta(x_i, y_i)$. If $q = \text{QUANTILE}(\{s_i\}_{i=1}^M, \ 1 - \alpha)$ (see Algorithm 1) is the empirical $\frac{\lceil (M+1)(1-\alpha) \rceil}{M}$-quantile of the set $\{s_i\}_{i=1}^M$ and $(x, y)$ is drawn exchangeably with $D_{cal}$, then $\Omega_\theta(x)$ has the marginal coverage guarantee*

$$1 - \alpha \leq \mathbb{P}_{x,y,D_{cal}}(y \in \Omega_\theta(x)) \leq 1 - \alpha + \frac{1}{M+1}.$$

We use split conformal prediction, rather than full conformal prediction, both for computational tractability and to avoid the problem of nonconvex uncertainty sets that can arise from the full conformal approach, as noted in Johnstone & Cox (2021). For the rest of this paper, we assume $\alpha \in [\frac{1}{M+1}, 1)$ so that $q = \text{QUANTILE}(\{s_i\}_{i=1}^M, \ 1 - \alpha) < \infty$ is finite. Thus, for appropriate choices of the score function $s_\theta$, the uncertainty set $\Omega_\theta(x)$ is not unbounded.

While the split conformal prediction procedure in Lemma 3.2 ensures that the uncertainty set $\Omega_\theta(x)$ satisfies $(1 - \alpha)$ coverage at inference time, this process does not address the question of *training* the uncertainty set $\Omega_\theta(x)$ (via the score function $s_\theta$) to ensure optimal task performance while maintaining coverage. In Section 3.3, we propose applying a separate differentiable conformal prediction procedure during training to address this challenge.

## 3.3  End-to-end training and calibration

Thus far, we have discussed how to calibrate an uncertainty set $\Omega_\theta(x)$ of the form (3) to ensure coverage, and we described two choices of score function $s_\theta$ parametrizing common box and ellipsoidal uncertainty sets. However, to ensure that the uncertainty sets $\Omega_\theta(x)$ *both* guarantee coverage *and* ensure optimal downstream task performance, it is necessary to design an end-to-end training methodology that can incorporate both desiderata in a fully differentiable manner. We propose such a methodology in Algorithm 1.

Our end-to-end training approach minimizes the empirical task loss $\ell(\theta) = \frac{1}{N} \sum_{i=1}^{N} \ell_i(\theta)$ using minibatch gradient descent, where $\ell_i(\theta) = f(x_i, y_i, z_\theta^\star(x_i))$. This requires differentiating through both the robust optimization problem as well as the conformal prediction step. The gradient of the task loss on a single instance is $\frac{d\ell_i}{d\theta} = \frac{\partial f}{\partial z}(x_i, y_i, z_\theta^\star(x_i)) \cdot \frac{\partial z_\theta^\star}{\partial \theta}(x_i)$, where $\frac{\partial z_\theta^\star}{\partial \theta}(x_i)$ is computed by differentiating through the Karush–Kuhn–Tucker (KKT) conditions of the convex reformulation of the optimization problem (1) (i.e., the problems (4), (5)) following the approach of Amos & Kolter (2017), under mild assumptions on the differentiability of $f$ and $g$. Note that the gradient of any convex optimization problem can be computed with respect to its parameters as such (Agrawal et al., 2019, Appendix B).

To include calibration during training, we assume that for every $(x, y)$ in our training set, $s_\theta(x, y)$ is differentiable w.r.t. $\theta$ almost everywhere; this assumption holds for common nonconformity score functions, including those used in this paper. We then adopt the conformal training approach (Stutz et al., 2022) in which a separate $q$ is chosen in each minibatch, as shown in Algorithm 1. The chosen $q$ depends on $\theta$ (through $s_\theta$), and $z_\theta^\star(x_i)$ depends on the chosen $q$. Therefore $\frac{\partial z_\theta^\star}{\partial \theta}$ involves calculating $\frac{\partial z_\theta^\star}{\partial q} \frac{\partial q}{\partial \theta}$, where $\frac{\partial q}{\partial \theta}$ requires differentiating through the empirical quantile function. Whereas Stutz et al. (2022) uses a smoothed approximate quantile function for calculating $q$, we find the smoothing unnecessary, as the gradient of the empirical quantile function is unique and well-defined almost everywhere. Importantly, our exact gradient is both more computationally efficient and simpler to implement than the smoothed approximate quantile approach. See Appendix D for more details.

After training has concluded and we have performed the final conformal calibration step, the resulting model enjoys the following theoretical guarantee on performance (*cf.* Sun et al. (2023), Proposition 1)).

**Proposition 3.3.** *Under the same assumptions as Lemma 3.2, the task loss satisfies the following bound with probability at least $1 - \alpha$ (over $x, y$, and the calibration set $D_{cal}$):*

$$f(x, y, z_\theta^\star(x)) \leq \left( \min_{z \in \mathbb{R}^p} \max_{\hat{y} \in \Omega_\theta(x)} f(x, \hat{y}, z) \text{ s.t. } g(x, z) \leq 0 \right). \tag{6}$$

*Proof.* When $y \in \Omega_\theta(x)$, the definition of $z_\theta^\star(x)$ (1) guarantees that (6) holds. By Lemma 3.2, $\mathbb{P}_{x, y, D_{cal}}(y \in \Omega_\theta(x)) \geq 1 - \alpha$. $\square$

We note that in practice, training models from scratch using only the task loss sometimes leads to poor local optima. Instead, in our experiments, we pretrain our models with a standard loss function (e.g., negative log-likelihood for the mean-variance predictor used in ellipsoidal uncertainty sets), and then we fine-tune the model using the task loss. This is a standard heuristic commonly used in the decision-focused learning and predict-then-optimize literature (Donti et al., 2017; Mandi et al., 2024). Appendix E provides more details on our training procedure.

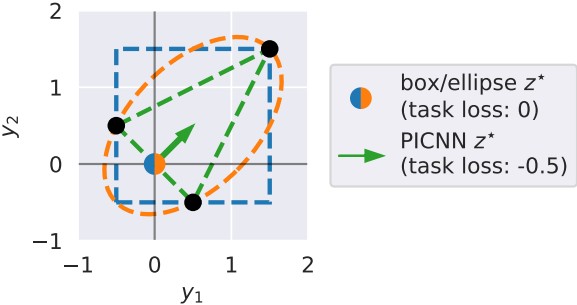

Figure 2: Consider a robust portfolio optimization problem with 2 assets, where $y \in \mathbb{R}^2$ is a random vector of asset returns, and the decision $z \in \mathbb{R}^2$ represents portfolio weights: $\max_z \min_{\hat{y} \in \Omega} -z^\top \hat{y}$ s.t. $z \geq 0$, $\mathbf{1}^\top z \leq 1$. Let the distribution of asset returns $y$ be uniform over 3 discrete points (black). The optimal box (blue), ellipse (orange), and PICNN (green) uncertainty sets are shown with their robust decision vectors $z^\star$. The flexibility of the PICNN uncertainty representation allows it to achieve the lowest expected task loss.

# 4 Representing General Convex Uncertainty Sets via PICNNs

The previous section discussed how to train calibrated box and ellipsoidal uncertainty sets end-to-end to optimize the downstream task loss. However, both box and ellipsoidal uncertainty sets have restrictive shapes which may yield suboptimal task performance. If $\Omega_\theta(x)$ could represent *any* arbitrary convex uncertainty set, this more expressive class would enable obtaining better task loss. Figure 2 illustrates an example where a general convex uncertainty set representation provides a clear advantage over box and ellipsoid uncertainty.

To this end, we propose to directly learn a partially-convex nonconformity score function $s_\theta : \mathbb{R}^m \times \mathbb{R}^n \to \mathbb{R}$ that is convex only in the second input vector. Fixing $x$, any $q$-sublevel set $\{\hat{y} \in \mathbb{R}^n \mid s_\theta(x, \hat{y}) \leq q\}$ of $s_\theta$ is a convex set, and likewise every family of convex sets can be expressed as the $q$-sublevel sets of some partially-convex function. To implement this approach, we are faced with two questions.

*1. How should we parametrize the score function $s_\theta$ so $\Omega_\theta(x)$ can approximate* arbitrary *convex sets?* A natural answer is to parametrize $s_\theta$ with a partially input-convex neural network (PICNN) (Amos et al., 2017), which can efficiently approximate any partially-convex function (Chen et al., 2019). We consider a PICNN defined as $s_\theta(x, y) = W_L \sigma_L + V_L y + b_L$, where

$$
\begin{aligned}
\sigma_0 &= \mathbf{0}, & u_0 &= x, & W_l &= \bar{W}_l \operatorname{diag}([\hat{W}_l u_l + w_l]_+) \\
u_{l+1} &= \operatorname{ReLU}(R_l u_l + r_l), & V_l &= \bar{V}_l \operatorname{diag}(\hat{V}_l u_l + v_l) & & (7) \\
\sigma_{l+1} &= \operatorname{ReLU}(W_l \sigma_l + V_l y + b_l), & b_l &= \bar{B}_l u_l + \bar{b}_l,
\end{aligned}
$$

with weights $\theta = (R_l, r_l, \bar{W}_l, \hat{W}_l, w_l, \bar{V}_l, \hat{V}_l, v_l, \bar{B}_l, \bar{b}_l)_{l=0}^L$. The matrices $\bar{W}_l$ are constrained to be entrywise nonnegative to ensure convexity of $s_\theta$ with respect to $y$. For ease of notation, we assume all hidden layers $\sigma_1, \ldots, \sigma_L$ have the same dimension $d$.

*2. Does the chosen parametrization of $\Omega_\theta(x)$ (via PICNNs) yield a tractable reformulation of the CRO problem* (1)*?* Fortunately, we show in the following theorem that the answer is yes.

**Theorem 4.1.** *Let $\Omega_\theta(x) = \{\hat{y} \in \mathbb{R}^n \mid s_\theta(x, \hat{y}) \leq q\}$, where $s_\theta$ is a PICNN as defined in* (7)*. Then, under Assumption 3.1, the CRO problem* (1) *with uncertainty set $\Omega_\theta(x)$ is equivalent to the following convex (and hence tractable) minimization problem:*

$$
\begin{aligned}
z_\theta^\star(x) = \arg\min_{z \in \mathbb{R}^p} \min_{\nu \in \mathbb{R}^{2Ld+1}} \quad & b(\theta, q)^\top \nu + \tilde{f}(x, z) \\
\text{s.t.} \quad & A(\theta)^\top \nu = \begin{bmatrix} F(x, z) \\ \mathbf{0} \end{bmatrix}, \ \nu \geq \mathbf{0}, \ g(x, z) \leq 0
\end{aligned}
\tag{8}
$$

*where $A(\theta) \in \mathbb{R}^{(2Ld+1) \times (n+Ld)}$ and $b(\theta, q) \in \mathbb{R}^{2Ld+1}$ are constructed from the weights $\theta$ of the PICNN* (7)*, and $b$ also depends on $q$.*

We prove Theorem 4.1 in Appendix C.3; the main idea is that when $\Omega_\theta(x)$ is a sublevel set of a PICNN, we can equivalently reformulate the inner maximization problem in (1) as a linear program and take the dual to yield a tractable minimization problem.

Since the PICNN uncertainty sets are of the same form as (3) and yield a tractable convex reformulation (8) of the CRO problem (1), we can apply the split conformal procedure detailed in Section 3.2 to choose $q \in \mathbb{R}$ and obtain coverage guarantees on $\Omega_\theta(x)$, and we can employ the same end-to-end training methodology from Section 3.3 to train calibrated uncertainties end-to-end using the downstream task loss. In some cases during training, the inner maximization problem of (1) with PICNN-parametrized uncertainty set may be unbounded (if $\Omega_\theta(x)$ is not compact) or infeasible (if the chosen $q$ is too small causing $\Omega_\theta(x)$ to be empty). This will lead, respectively, to an infeasible or unbounded equivalent problem (8). We can avoid this concern by adjusting the PICNN architecture to ensure its sublevel sets are compact and by suitably increasing $q$ when needed to ensure $\Omega_\theta(x)$ is never empty. Such modifications do not change the general form of the problem (8) and preserve the marginal coverage guarantee for the uncertainty set $\Omega_\theta(x)$; see Appendix C.4 for details.

## 5 Experiments

In this section, we present experimental results for our E2E method against several ETO baselines. Code to reproduce our results is available on GitHub.[3]

### 5.1 Problem descriptions

We consider two tasks: price forecasting for battery storage operation and portfolio optimization. Their task loss functions and constraints satisfy Assumption 3.1.

**Price forecasting for battery storage.** This problem comes from Donti et al. (2017), where a grid-scale battery operator predicts electricity prices $y \in \mathbb{R}^T$ over a $T$-step horizon and uses the predicted prices to decide a battery charge/discharge schedule for price arbitrage. The input features $x$ include the past day's prices and temperature, the next day's energy load forecast and temperature forecast, binary indicators of weekends or holidays, and yearly sinusoidal features. The operator decides how much to charge ($z^{\text{in}} \in \mathbb{R}^T$) or discharge ($z^{\text{out}} \in \mathbb{R}^T$) the battery, which changes the battery's state of charge ($z^{\text{state}} \in \mathbb{R}^T$). The battery has capacity $B$, charging efficiency $\gamma$, and maximum charging/discharging rates $c^{\text{in}}$ and $c^{\text{out}}$. The task loss function represents the multiple objectives of maximizing profit, flexibility to participate in other markets by keeping the battery near half its capacity (with weight $\lambda$), and battery health by discouraging rapid charging/discharging (with weight $\epsilon$):

$$f(y, z) = \sum_{t=1}^{T} y_t (z^{\text{in}} - z^{\text{out}})_t + \lambda \left\| z^{\text{state}} - \frac{B}{2} \mathbf{1} \right\|^2 + \epsilon \left\| z^{\text{in}} \right\|^2 + \epsilon \left\| z^{\text{out}} \right\|^2.$$

The constraints are, for all $t = 1, \ldots, T$,

$$z_0^{\text{state}} = B/2, \quad z_t^{\text{state}} = z_{t-1}^{\text{state}} - z_t^{\text{out}} + \gamma z_t^{\text{in}},$$
$$0 \le z^{\text{in}} \le c^{\text{in}}, \quad 0 \le z^{\text{out}} \le c^{\text{out}}, \quad 0 \le z_t^{\text{state}} \le B.$$

Following Donti et al. (2017), we set $T = 24$, $B = 1$, $\gamma = 0.9$, $c^{\text{in}} = 0.5$, $c^{\text{out}} = 0.2$, $\lambda = 0.1$, and $\epsilon = 0.05$.

**Portfolio optimization.** We adopt the portfolio optimization setting and synthetic dataset from Chenreddy & Delage (2024), where the prediction targets $y \in \mathbb{R}^n$ are the returns of a set of $n$ securities, and the decision $z \in \mathbb{R}^n$ sets portfolio weights. The task loss is $f(y, z) = -y^\top z$, with constraints $z \ge 0$, $\mathbf{1}^\top z = 1$. The synthetic dataset consists of $(x, y) \in \mathbb{R}^{2 \times 2}$ drawn from a mixture of three 4-D multivariate normal distributions. We provide data details in Appendix E.2 and experimental results in Appendix A.2. The results are similar to those for battery storage, except that portfolio optimization is a lower-dimensional and easier problem.

---

[3]https://github.com/chrisyeh96/e2e-conformal

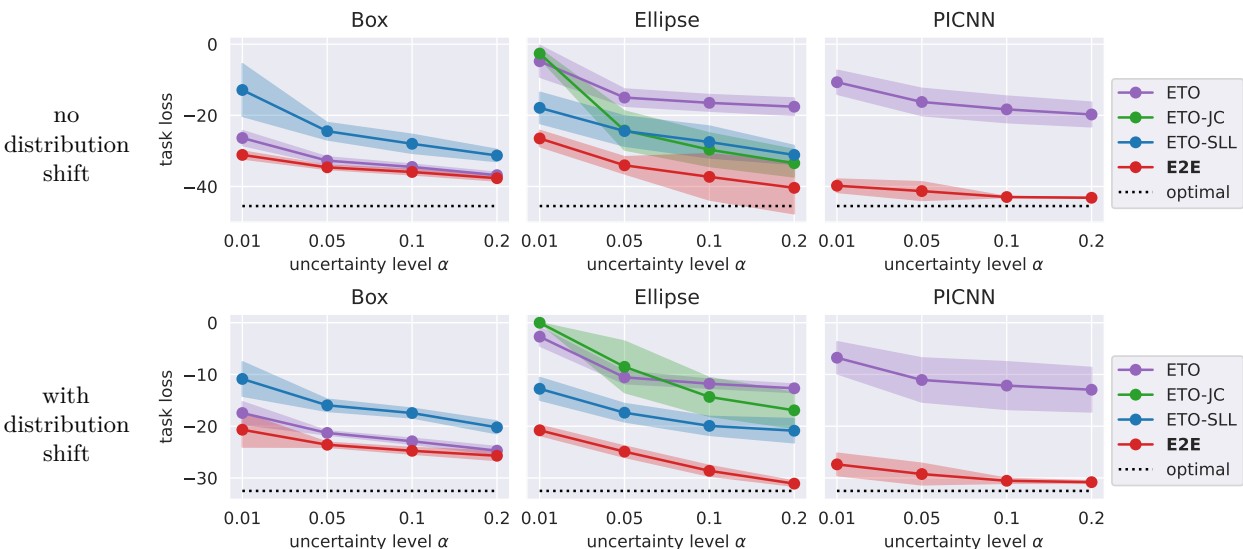

Figure 3: Task loss performance (mean $\pm 1$ stddev across 10 runs) for the battery storage problem with no distribution shift (top) and with distribution shift (bottom). Lower values are better.

## 5.2 Baseline methods

We implemented several "estimate-then-optimize" (ETO) baselines, listed below, to compare against our end-to-end (E2E) method. These two-stage ETO baselines are trained using task-agnostic losses such as pinball loss or negative log-likelihood (NLL). To ensure a fair comparison against our E2E method, we also apply conformal calibration to each ETO method after training to satisfy coverage.

- **ETO** denotes models with identical neural network architectures to our E2E models, differing only in the loss function during training. The box uncertainty **ETO** model is trained with pinball loss to predict the $\frac{\alpha}{2}$ and $1 - \frac{\alpha}{2}$ quantiles. The ellipsoidal uncertainty **ETO** model is trained with a multivariate normal negative log-likelihood loss. For the PICNN **ETO** model, we train $s_\theta$ using a negative log-likelihood loss by interpreting $s_\theta$ as an energy function—i.e., $\hat{\mathcal{P}}_\theta(y \mid x) \propto \exp(-s_\theta(x,y))$—yielding the loss $\mathrm{NLL}(\theta) = \ln s_\theta(x,y) + \ln Z_\theta(x)$, where $Z_\theta(x) = \int_{\tilde{y} \in \mathbb{R}^n} \exp(-s_\theta(x,\tilde{y})) \, \mathrm{d}\tilde{y}$, following the approach of Lin & Ba (2023). More details of the **ETO** models are given in Appendix E.
- **ETO-SLL** is our implementation of the box and ellipsoid uncertainty ETO methods from Sun et al. (2023). Unlike **ETO**, **ETO-SLL** first trains a point estimate model (without uncertainty) with mean-squared error loss. Then, **ETO-SLL** box and ellipsoidal uncertainty sets are derived from training a separate quantile regressor using pinball loss to predict the $(1 - \alpha)$-quantiles of absolute residuals or $\ell_2$-norm of residuals of the point estimate. Unlike **ETO** which can learn ellipsoidal uncertainty sets with different covariance matrices for each input $x$, the **ETO-SLL** ellipsoidal uncertainty sets all share the same covariance matrix (up to scale).
- **ETO-JC** is our implementation of the ellipsoid uncertainty ETO method by Johnstone & Cox (2021). Like **ETO-SLL**, **ETO-JC** also first trains a point estimate model (without uncertainty) with mean-squared error loss. **ETO-JC** uses the same covariance matrix (with the same scale) for each input $x$.

## 5.3 Battery storage problem results

Figure 3 (top) compares task loss performance for different uncertainty levels ($\alpha \in \{.01, .05, .1, .2\}$) and the different uncertainty set representations for the ETO baselines against our proposed E2E methodology on the battery storage problem with no distribution shift. Our E2E approach consistently yields improved performance over all ETO baselines, for all three uncertainty set parametrizations, and over all tested uncertainty levels $\alpha$. Moreover, the PICNN uncertainty representation, when trained end-to-end, provides up to 42% relative improvement in performance over the best ETO box uncertainty set and up to 209%

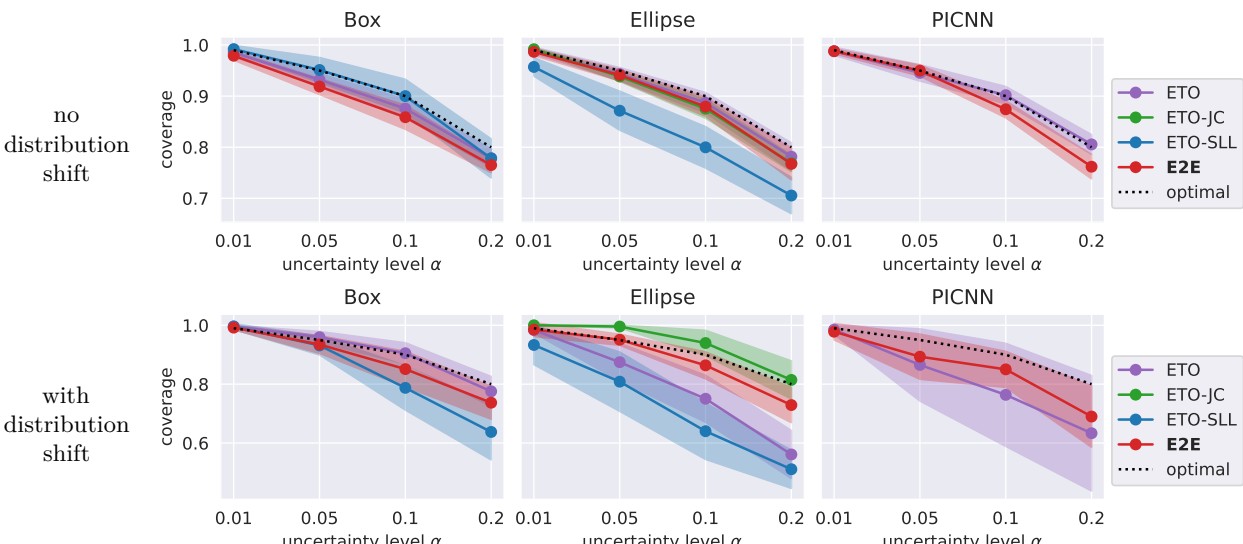

Figure 4: Coverage (mean $\pm 1$ stddev across 10 runs) for the battery storage problem with no distribution shift (top) and with distribution shift (bottom). The dotted black line indicates the target coverage level $1 - \alpha$. Our E2E models achieve similar coverage to the ETO baselines, confirming that the lower task loss of our E2E models does not come at the expense of worse coverage.

relative improvement over the best ETO ellipse uncertainty set. We also show the corresponding coverage obtained by the learned uncertainty sets in Figure 4 (top); all models obtain coverage close to the target level, confirming that the improvements in task loss performance from our E2E approach do not come at the cost of worse coverage.

In Appendix A, we present additional results on the battery storage problem, including value-at-risk (VaR) and conditional value-at-risk (CVaR) metrics which show that the E2E approach consistently outperforms the ETO baselines on tail risk as well. In addition, Table A1 shows the per-epoch computational time required by our E2E approach compared with the two-stage baselines. While E2E requires additional training time, the training time is dominated by time spent solving the conditional robust optimization problem (1) for every training example at every epoch. The development of GPU-accelerated solvers (Blin, 2024) may help significantly speed up the optimization, though we do not explore GPU-accelerated optimization in our implementation.

## 5.4 Performance under distribution shift

The aforementioned results were produced without distribution shift, where our training and test sets were sampled uniformly at random, thus ensuring exchangeability and guaranteeing marginal coverage. In this section, we evaluate our method on the more realistic setting with distribution shift by splitting our data temporally; our models are trained on the first 80% of days and evaluated on the last 20% of days. Figure A4 plots the underlying time series data and visually highlights the distribution shift. Figures 3 (bottom) and 4 (bottom) mirror Figures 3 (top) and 4 (top), except that there is now distribution shift. We again find that our E2E approach consistently yields improved performance over all ETO baselines, for all three uncertainty set parametrizations, and for all tested uncertainty levels $\alpha$. Likewise, the PICNN uncertainties, when trained end-to-end, improve on the performance offered by box and ellipsoidal uncertainty. We unsurprisingly find that, under distribution shift, the models do not provide the same level of coverage guaranteed in the i.i.d. case, as the exchangeability assumption needed for conformal prediction no longer holds. The ellipsoidal and PICNN models tend to provide worse coverage than the box uncertainty, which we believe reflects how ellipsoidal and PICNN uncertainty sets offer greater representational power, and thus might be fitting too closely to the pre-shift distribution, which impacts robustness under distribution shift. Devising methods to

anticipate distribution shift when training these more expressive models, and in particular the PICNN-based uncertainty, remains an interesting avenue for future work.

## 6 Conclusion

In this work, we develop the first end-to-end methodology for training predictive models with uncertainty estimates (with calibration enforced differentiably throughout training) that are utilized in downstream conditional robust optimization problems. We demonstrate an approach utilizing partially input-convex neural networks (PICNNs) to represent general convex uncertainty regions, and we perform extensive experiments on a battery storage application and a portfolio optimization task. Whereas prior works on two-stage estimate-then-optimize approaches emphasized "the convenience brought by the disentanglement of the prediction and the uncertainty calibration" (Sun et al., 2023), our results highlight that such "convenience" comes at a substantial cost; our end-to-end approach, combined with the expressiveness of the PICNN representation, has clear performance gains over the traditional two-stage methods.

A number of interesting directions for future work on learning decision-aware uncertainty in an end-to-end manner remain. First, while our PICNN-based uncertainty set representation allows the parametrization of general convex uncertainty sets, future work may explore *nonconvex* uncertainty regions. Doing so may require eschewing the analytical methods for differentiating through convex optimization problems and instead use, e.g., policy gradient methods for passing gradients through general stochastic and robust optimization problems. Second, developing end-to-end methods to target conditional calibration (as opposed to marginal calibration) remains an active research direction. Finally, one may explore other types of constraints besides uncertainty set-based robustness, such as value-at-risk (VaR) or conditional value-at-risk (CVaR) constraints.

### Acknowledgments

We thank Priya Donti for helpful discussions. The authors acknowledge support from an NSF Graduate Research Fellowship (DGE-2139433); NSF Grants CNS-2146814, CPS-2136197, CNS-2106403, and NGSDI-2105648; gifts from Amazon, OpenAI, and Latitude AI; and the Caltech Resnick Sustainability Institute.

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

Table A1: Time per epoch of pretraining, optimization, and end-to-end training, measured in seconds (wall-clock time). Values are mean $\pm 1$ stddev across 10 epochs. Lower values are better.

|  | pretrain | optimize (train+val) | E2E training |
|---|---|---|---|
| Box | $0.03 \pm 0.01$ | $4.29 \pm 0.07$ | $6.34 \pm 0.32$ |
| Ellipse | $0.03 \pm 0.01$ | $7.79 \pm 0.06$ | $7.12 \pm 1.28$ |
| PICNN | $4.20 \pm 0.20$ | $50.49 \pm 0.19$ | $33.24 \pm 0.25$ |

# A   Appendix: Additional experimental results

## A.1   Experimental results: Battery storage

In Figures A1 and A2, we plot the $(1-\alpha)$-value-at-risk (VaR) and $(1-\alpha)$-conditional value-at-risk (CVaR) (Uryasev & Rockafellar, 2001) of the task loss obtained via our method on the battery storage problem to evaluate model robustness. In particular, for each target coverage level $1-\alpha$, we evaluate $\text{VaR}^{1-\alpha}$[task loss] and $\text{CVaR}^{1-\alpha}$[task loss] for both the ETO baseline(s) and our E2E methodologies across box, ellipsoid, and PICNN uncertainty sets. In general, it is clear that our E2E methodology uniformly improves over ETO, and while the performance of E2E ellipsoid is close to that of E2E PICNN, the latter appears to perform better for certain $\alpha$. These results highlight that our methodology improves not only the average task loss, but robustness more generally, even when compared with the robust ETO baselines.

The optimal task losses shown in black dotted lines in Figures 3, A1 and A2 are the lowest achievable task loss on the test set given perfect knowledge of the target $y$. The optimal task loss is calculated for each example $(x, y)$ in the test set as $f(x, y, z_{\text{opt}}^\star)$ where

$$z_{\text{opt}}^\star = \underset{z \in \mathbb{R}^p}{\arg\min} \, f(x, y, z) \text{ s.t. } g(x, z) \leq 0.$$

**Performance.**   In Table A1, we report the wall-clock time for pretraining, optimization, and end-to-end training. These times were measured on a machine with $2\times$ AMD EPYC 7513 32-Core Processors, 1TiB RAM, and 4 NVIDIA A100 GPUs (although only 1 of the GPUs was used in these experiments). The "pretrain" column gives the time per epoch of training with the standard loss function (pinball loss for Box, negative log-likelihood loss for Ellipse and PICNN). The "optimize" column gives the time needed to compute the decision $z_\theta^\star(x)$ given a pretrained model. The "E2E" column gives the time per epoch of E2E training, which requires computing $z_\theta^\star(x)$ for each training and calibration example. Evidently, the bulk of the time spent on E2E training comes from computing the decision $z_\theta^\star(x)$; the additional overhead from computing the gradient through the KKT conditions of the optimization problem is much less than solving the optimization problem itself.

We note that in theory, E2E times should always be higher than the optimization time. The optimization time accounts for the time required to compute $z_\theta^\star(x)$, and E2E training additional accounts for the time required to compute the gradient $\frac{\partial}{\partial\theta} z_\theta^\star(x)$. However, in practice, the optimization time depends heavily on the choice of numerical solver. In our implementation, we use the default `cvxpy` solver (Clarabel) for the optimization step in ETO, whereas we use the default `cvxpylayers` solver (SCS) during E2E training. In the case of the Ellipse and PICNN uncertainty sets, the SCS solver is generally faster than Clarabel; hence, Table A1 counterintuitively reports lower E2E training times for Ellipse and PICNN than optimization times.

We further note that the optimization and E2E training times reflect our particular implementation, but significantly faster times are likely possible. In particular, we solve all of the optimization problems (in both the ETO optimization step and during E2E training) using CPU-based solvers; however, recently developed GPU-accelerated solvers may be able to reduce the optimization time required by orders of magnitude, especially when solving a batch of problems with the same structure (Blin, 2024).

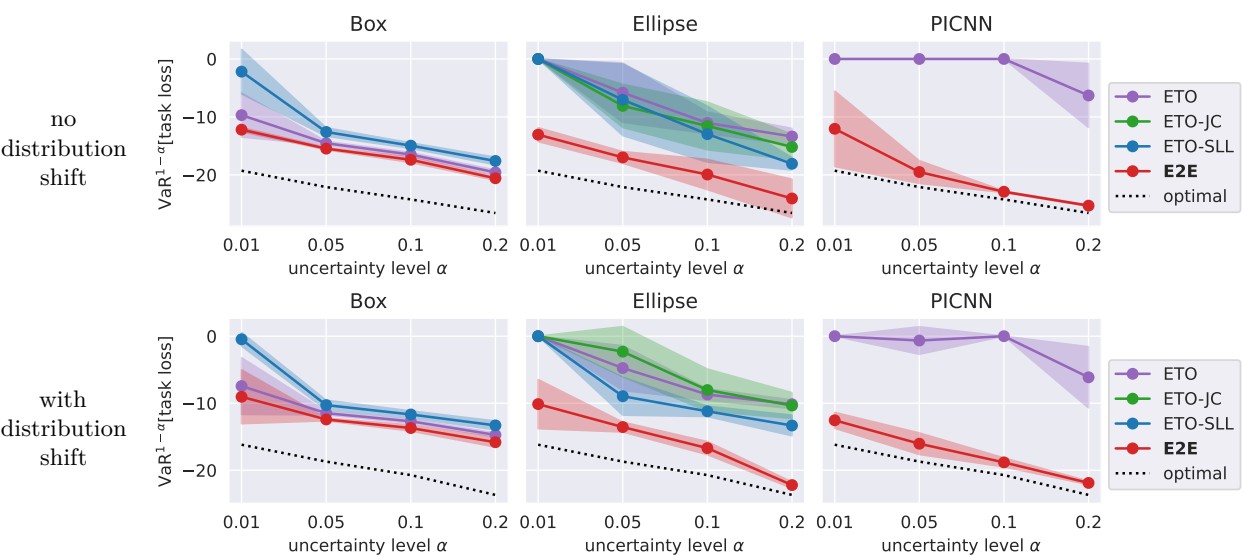

Figure A1: Similar to Figure 3, except that the value plotted is the $\mathrm{VaR}^{1-\alpha}$[task loss] (mean $\pm 1$ stddev across 10 runs) on the test set for the battery storage problem with no distribution shift (top) and with distribution shift (bottom). Lower values are better.

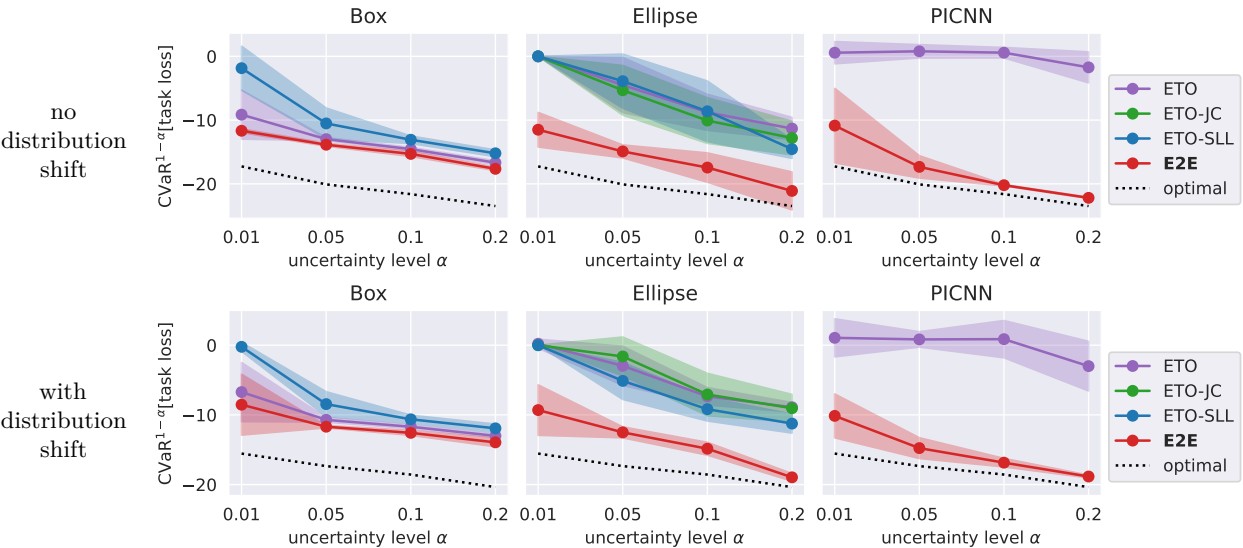

Figure A2: Similar to Figure 3, except that the value plotted is the $\mathrm{CVaR}^{1-\alpha}$[task loss] (mean $\pm 1$ stddev across 10 runs) on the test set for the battery storage problem with no distribution shift (top) and with distribution shift (bottom). Lower values are better.

## A.2 Experimental results: Portfolio optimization

Tables A2 and A3 show the task loss and coverage results for the portfolio optimization problem. We again find that our E2E approach generally improves upon the ETO baselines at all uncertainty levels $\alpha$, with the exception of box uncertainty, where all the methods achieve similar performance. Our PICNN-based uncertainty representation, when learned end-to-end, performs better than box uncertainty and comparably with ellipse uncertainty. The similarity in performance between E2E ellipsoidal uncertainty and E2E PICNN uncertainty is likely due to the underlying aleatoric uncertainty (i.e., the uncertainty in $\mathcal{P}(y \mid x)$) generally taking an ellipsoidal shape—the conditional distribution $\mathcal{P}(y \mid x)$ is a Gaussian mixture model, and it tends

Table A2: Task loss performance (mean $\pm$ 1 stddev across 10 runs) for the portfolio optimization problem. Lower values are better, and the best performance for each uncertainty-level $\alpha$ is highlighted. The results show that our E2E methods consistently outperform the ETO baselines.

| | | uncertainty level $\alpha$ | | | |
| | | 0.01 | 0.05 | 0.1 | 0.2 |
|---|---|---|---|---|---|
| ETO | Box | $-1.16 \pm 0.42$ | $-1.37 \pm 0.12$ | $-1.39 \pm 0.13$ | $-1.41 \pm 0.12$ |
| ETO | Ellipse | $-1.09 \pm 0.12$ | $-1.24 \pm 0.11$ | $-1.29 \pm 0.10$ | $-1.33 \pm 0.10$ |
| ETO | PICNN | $-0.95 \pm 0.24$ | $-1.11 \pm 0.24$ | $-1.20 \pm 0.22$ | $-1.31 \pm 0.16$ |
| ETO-SLL | Box | $-1.41 \pm 0.13$ | $-1.42 \pm 0.12$ | $-1.42 \pm 0.12$ | $-1.44 \pm 0.11$ |
| ETO-SLL | Ellipse | $-1.12 \pm 0.22$ | $-1.37 \pm 0.12$ | $-1.40 \pm 0.12$ | $-1.43 \pm 0.12$ |
| ETO-JC | Ellipse | $-1.16 \pm 0.17$ | $-1.40 \pm 0.11$ | $-1.42 \pm 0.11$ | $-1.44 \pm 0.11$ |
| E2E | Box | $-1.21 \pm 0.44$ | $-1.40 \pm 0.14$ | $-1.43 \pm 0.11$ | $-1.43 \pm 0.10$ |
| E2E | Ellipse | $\mathbf{-1.48} \pm 0.12$ | $-1.47 \pm 0.11$ | $\mathbf{-1.48} \pm 0.11$ | $\mathbf{-1.47} \pm 0.11$ |
| E2E | PICNN | $-1.45 \pm 0.14$ | $\mathbf{-1.48} \pm 0.10$ | $\mathbf{-1.48} \pm 0.10$ | $\mathbf{-1.47} \pm 0.11$ |

Table A3: Coverage (mean $\pm 1$ stddev across 10 runs) for the portfolio optimization problem. Our E2E models achieve similar coverage to the ETO baselines, confirming that the lower task loss of our E2E models does not come at the expense of worse coverage.

| | | uncertainty level $\alpha$ | | | |
| | | 0.01 | 0.05 | 0.1 | 0.2 |
|---|---|---|---|---|---|
| ETO | Box | $.984 \pm .007$ | $.947 \pm .017$ | $.902 \pm .017$ | $.786 \pm .020$ |
| ETO | Ellipse | $.988 \pm .004$ | $.944 \pm .020$ | $.894 \pm .022$ | $.794 \pm .027$ |
| ETO | PICNN | $.989 \pm .006$ | $.949 \pm .014$ | $.901 \pm .019$ | $.801 \pm .034$ |
| ETO-SLL | Box | $.985 \pm .012$ | $.945 \pm .021$ | $.885 \pm .030$ | $.796 \pm .029$ |
| ETO-SLL | Ellipse | $.989 \pm .011$ | $.945 \pm .024$ | $.885 \pm .039$ | $.795 \pm .030$ |
| ETO-JC | Ellipse | $.991 \pm .006$ | $.953 \pm .017$ | $.902 \pm .026$ | $.796 \pm .026$ |
| E2E | Box | $.989 \pm .006$ | $.949 \pm .012$ | $.903 \pm .016$ | $.785 \pm .019$ |
| E2E | Ellipse | $.992 \pm .006$ | $.954 \pm .010$ | $.903 \pm .022$ | $.798 \pm .022$ |
| E2E | PICNN | $.993 \pm .002$ | $.953 \pm .010$ | $.912 \pm .017$ | $.798 \pm .024$ |

to have a dominant mode (e.g., see Figure A3). In terms of coverage, we find that all the models and training methodologies obtain coverage very close to the target level, confirming that the improvements in task loss performance from our E2E approach do not come at the cost of worse coverage.

Because the conditional distribution $\mathcal{P}(y \mid x)$ for the portfolio optimization problem is 2-dimensional, we can visualize the conditional distribution as well as the uncertainty sets estimated by our models. Figure A3 plots the conditional density for input $x = \begin{bmatrix} -1.167 & 0.024 \end{bmatrix}^\top$, along with the $\alpha = 0.1$ uncertainty sets $\Omega_\theta(x)$ and the resulting decision vectors $z_\theta^\star(x)$ for each uncertainty set parametrization. Uncertainty sets and decision vectors from both ETO and E2E models are shown in different colors. The key takeaway from this figure is that smaller uncertainty sets (which is what ETO training tends to produce) do not always result in lower task loss. Furthermore, the more flexible parametrization of the PICNN allows it to learn uncertainty set shapes that may be more amenable to the downstream robust decision task than box or ellipsoidal uncertainty, even if the resulting uncertainty set has a larger or odder shape.

# B   Appendix: Related Work

**"Task-based" or "decision-focused" learning.**   The notion of "task-based" end-to-end model learning was introduced by Donti et al. (2017), which proposed to train machine learning models in an end-to-end fashion to minimize a downstream stochastic optimization objective. To achieve this, the authors

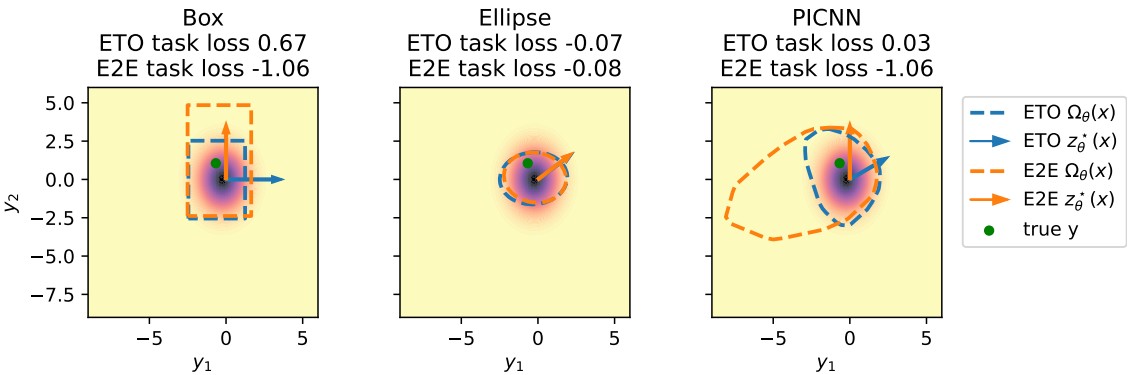

Figure A3: This figure plots the density of the conditional distribution $\mathcal{P}(y \mid x)$ for $x = \begin{bmatrix} -1.167 & 0.024 \end{bmatrix}^\top$ from the portfolio optimization problem, with darker colors indicating higher density. Also plotted are the $\alpha = 0.1$ uncertainty sets $\Omega_\theta(x)$ (dashed lines) and the resulting decision vectors $z_\theta^\star(x)$ (arrows) for each uncertainty set parametrization. Results for `ETO` models are shown in blue, whereas results for E2E are shown in orange. The "true" $y$ sampled from $\mathcal{P}(y \mid x)$ is drawn in green, and the task loss for this example is computed using this $y$. The decision vectors have been artificially scaled larger to be easier to see.

backpropagate gradients through a stochastic optimization problem, which is made possible for various types of convex optimization problems via the implicit function theorem (Donti et al., 2017; Amos & Kolter, 2017; Agrawal et al., 2019). However, Donti et al. (2017) does not train the model to estimate uncertainty and thereby does not provide any explicit guarantees on robustness on their decisions to uncertainty. Our framework improves upon this baseline by yielding calibrated uncertainty sets which can then be used to obtain robust decisions.

Another line of related works relies on surrogate loss functions to approximate the gradient of the downstream task loss, typically in settings where the exact gradient does not exist or is otherwise hard to compute. For example, the "Smart Predict, then Optimize" approach (Elmachtoub & Grigas, 2022) specifically considers differentiating the solution of linear programs, whereas Wilder et al. (2019) differentiates the solution of discrete (combinatorial) optimization problems. As with Donti et al. (2017), though, these approaches focus only on average task loss without any estimation of model uncertainty, thereby lacking explicit guarantees on the robustness of the resulting decisions. A recent survey by Mandi et al. (2024) provides a comprehensive review of existing decision-focused learning approaches.

**Uncertainty Quantification.** Various designs for deep learning regression models that provide uncertainty estimates have been proposed in the literature, including Bayesian neural networks (Blundell et al., 2015; Gal & Ghahramani, 2016), Gaussian process regression and deep kernel learning (Rasmussen & Williams, 2005; Wilson et al., 2016; Liu et al., 2020), ensembles of models (Lakshminarayanan et al., 2017), and quantile regression (Romano et al., 2019), among other techniques. These methods typically only provide heuristic uncertainty estimates that are not necessarily well-calibrated (Nado et al., 2022).

Post-hoc methods such as isotonic regression (Kuleshov et al., 2018) or conformal prediction (Shafer & Vovk, 2008) may be used to calibrate the uncertainty outputs of deep learning models. These calibration methods generally treat the model as a black box and scale predicted uncertainty levels so that they are calibrated on a held-out calibration set. Isotonic regression guarantees calibrated outputs in the limit of infinite data, whereas conformal methods provide probabilistic, finite-sample calibration guarantees when the calibration set is exchangeable (e.g., drawn i.i.d. from the same distribution) with test data. These calibration methods are generally not included in the model training procedure because they involve non-differentiable operators, such as sorting. However, recent works have proposed differentiable losses (Einbinder et al., 2022; Stutz et al., 2022) that approximate the conformal prediction procedure during training and thus allow end-to-end training of models to output more calibrated uncertainty. As approximations, these methods lose the marginal coverage guarantees that true conformal methods provide. However, such guarantees can be recovered at test time by replacing the approximations with true conformal prediction.

**Robust and stochastic optimization.** The optimization community has proposed a number of techniques over the years to improve robust decision-making under uncertainty, including stochastic, risk-sensitive, chance-constrained, distributionally robust, and robust optimization (e.g., Ben-Tal et al. (2009); Shapiro et al. (2009); Nemirovski & Shapiro (2007); Rahimian & Mehrotra (2019)). These techniques have been applied to a wide range of applications, including energy systems operation (Zheng et al., 2015; Ndrio et al., 2021; Dvorkin, 2020; Zhong et al., 2021; Poolla et al., 2021; Warrington et al., 2012; Bertsimas et al., 2013; Christianson et al., 2022) and portfolio optimization (Gregory et al., 2011; Quaranta & Zaffaroni, 2008; Bertsimas et al., 2018). In these works, the robust and stochastic optimization methods enable selecting decisions (grid resource dispatches or portfolio allocations) in a manner that is aware of uncertainty, e.g., so an energy system operator can ensure that sufficient generation is available to meet demand even on a cloudy day without much solar generation. Typically, however, the construction of uncertainty sets, estimated probability distributions over uncertain parameters, or ambiguity sets over distributions takes place offline and is unconnected to the eventual decision-making task. Thus, our proposed end-to-end approach allows for simultaneous calibration of uncertainty sets with optimal decision-making.

## C Appendix: Maximizing over the uncertainty set

We consider robust optimization problems of the form

$$\min_{z \in \mathbb{R}^p} \max_{\hat{y} \in \mathbb{R}^n} \; \hat{y}^\top F z + \tilde{f}(x, z) \qquad \text{s.t.} \qquad \hat{y} \in \Omega(x), \quad g(x, z) \leq 0.$$

For fixed $z$, the inner maximization problem is

$$\max_{\hat{y} \in \mathbb{R}^n} \; \hat{y}^\top F z \qquad \text{s.t.} \qquad \hat{y} \in \Omega(x),$$

which we analyze in the more abstract form

$$\max_{y \in \mathbb{R}^n} c^\top y \qquad \text{s.t.} \qquad y \in \Omega$$

for arbitrary $c \in \mathbb{R}^n \setminus \{0\}$. The subsections of this appendix derive the dual form of this maximization problem for specific representations of the uncertainty set $\Omega$.

Suppose $y$ is standardized or whitened by an affine transformation with $\mu \in \mathbb{R}^n$ and invertible matrix $W \in \mathbb{R}^{n \times n}$

$$y_{\text{transformed}} = W^{-1}(y - \mu)$$

so that $\Omega$ is an uncertainty set on the transformed $y_{\text{transformed}}$. Then, the original primal objective can be recovered as

$$c^\top y = c^\top (W y_{\text{transformed}} + \mu) = (Wc)^\top y_{\text{transformed}} + c^\top \mu.$$

In our experiments, we use element-wise standardization of $y$ by setting $W = \text{diag}(y_{\text{std}})$, where $y_{\text{std}} \in \mathbb{R}^n$ is the element-wise standard-deviation of $y$.

### C.1 Maximizing over a box constraint

Let $[\underline{y}, \overline{y}] \subset \mathbb{R}^n$ be a box uncertainty set for $y \in \mathbb{R}^n$. Then, for any vector $c \in \mathbb{R}^n$, the primal linear program

$$\max_{y \in \mathbb{R}^n} c^\top y \qquad \text{s.t.} \qquad \underline{y} \leq y \leq \overline{y}$$

has dual problem

$$\min_{\nu \in \mathbb{R}^{2n}} \begin{bmatrix} \overline{y}^\top & -\underline{y}^\top \end{bmatrix} \nu \qquad \text{s.t.} \qquad \begin{bmatrix} I_n & -I_n \end{bmatrix} \nu = c, \quad \nu \geq \mathbf{0},$$

which can also be equivalently written as

$$\min_{\nu \in \mathbb{R}^n} (\overline{y} - \underline{y})^\top \nu + \underline{y}^\top c \qquad \text{s.t.} \qquad \nu \geq \mathbf{0}, \quad \nu - c \geq \mathbf{0}.$$

Since strong duality always holds for linear programs, the optimal values of the primal and dual problems will be equal so long as one of the problems is feasible, e.g., so long as the box $[\underline{y}, \overline{y}]$ is nonempty. We can thus incorporate this dual problem into the outer minimization of (1) to yield the non-robust form (4).

## C.2 Maximizing over an ellipsoid

For any $c \in \mathbb{R}^n \setminus \{0\}$, $\Sigma \in \mathbb{S}_{++}^n$, and $q > 0$, the primal quadratically constrained linear program (QCLP)

$$\max_{y \in \mathbb{R}^n} \ c^\top y \qquad \text{s.t.} \qquad (y - \mu)^\top \Sigma^{-1} (y - \mu) \leq q$$

has dual problem

$$\min_{\nu \in \mathbb{R}} \ \frac{1}{4\nu} c^\top \Sigma c + \mu^\top c + \nu q \qquad \text{s.t.} \quad \nu \geq 0.$$

By Slater's condition, strong duality holds by virtue of the assumption that $q > 0$ (which implies strict feasibility of the primal problem), and thus the primal and dual problems have the same optimal value. Moreover, since $\Sigma$ is positive definite and $q > 0$, this problem has a unique optimal solution at $\nu^\star = \frac{1}{2\sqrt{q}} \|L^\top c\|_2$, where $L$ is the unique lower-triangular Cholesky factor of $\Sigma$ (i.e., $\Sigma = LL^\top$). Substituting $\nu^\star$ into the dual problem yields

$$\sqrt{q} \left\| L^\top c \right\|_2 + \mu^\top c.$$

Plugging this into (1) yields the non-robust form (5).

We write the dual objective in terms of the Cholesky factor $L$ because our predictive models for ellipsoidal uncertainty directly output the entries of $L$ (see Appendix E). Note, however, that the dual problem solution can be equivalently written in terms of the square-root of $\Sigma$, because

$$\left\| L^\top c \right\|_2^2 = c^\top L L^\top c = c^\top \Sigma c = c^\top \Sigma^{1/2} \Sigma^{1/2} c = \left\| \Sigma^{1/2} c \right\|_2^2.$$

## C.3 Proof of Theorem 4.1: Maximizing over the sublevel set of a PICNN

Let $s_\theta : \mathbb{R}^m \times \mathbb{R}^n \to \mathbb{R}$ be a partially input-convex neural network (PICNN) with ReLU activations as described in (7), so that $s_\theta(x, y)$ is convex in $y$. Suppose that all the hidden layers have the same dimension $d$ (i.e., $\forall l = 0, \ldots, L-1$: $W_l \in \mathbb{R}^{d \times d}$, $V_l \in \mathbb{R}^{d \times n}$, $b_l \in \mathbb{R}^d$), and the final layer $L$ has $W_L \in \mathbb{R}^{1 \times d}$, $V_L \in \mathbb{R}^{1 \times n}$, $b_L \in \mathbb{R}$. Let $c \in \mathbb{R}^n$ be any vector. Then, the optimization problem

$$\max_{y \in \mathbb{R}^n} \ c^\top y \qquad \text{s.t.} \qquad s_\theta(x, y) \leq q \tag{9}$$

can be equivalently written as

$$\max_{y \in \mathbb{R}^n, \ \sigma_1, \ldots, \sigma_L \in \mathbb{R}^d} \quad c^\top y \tag{10a}$$

$$\text{s.t.} \quad \sigma_l \geq \mathbf{0}_d \qquad \qquad \forall l = 1, \ldots, L \tag{10b}$$

$$\sigma_{l+1} \geq W_l \sigma_l + V_l y + b_l \qquad \forall l = 0, \ldots, L-1 \tag{10c}$$

$$W_L \sigma_L + V_L y + b_L \leq q. \tag{10d}$$

To see that this is the case, first note that (10) is a relaxed form of (9), obtained by replacing the equalities $\sigma_{l+1} = \text{ReLU}(W_l \sigma_l + V_l y + b_l)$ in the definition of the PICNN (7) with the two separate inequalities $\sigma_{l+1} \geq \mathbf{0}_d$ and $\sigma_{l+1} \geq W_l \sigma_l + V_l y + b_l$ for each $l = 0, \ldots, L-1$. As such, the optimal value of (10) is no less than that of (9). However, given an optimal solution $y, \sigma_1, \ldots, \sigma_L$ to (10), it is possible to obtain another feasible solution $y, \hat\sigma_1, \ldots, \hat\sigma_L$ with the same optimal objective value by iteratively decreasing each component of $\sigma_l$ until one of the two inequality constraints (10b), (10c) is tight, beginning at $l = 1$ and incrementing $l$ once all entries of $\sigma_l$ cannot be decreased further. This procedure of decreasing the entries in each $\sigma_l$ will maintain problem feasibility, since the weight matrices $W_l$ are all assumed to be entrywise nonnegative in the PICNN construction; in particular, this procedure will not increase the left-hand side of (10d). Moreover, since one of the two constraints (10b), (10c) will hold for each entry of each $\hat\sigma_l$, this immediately implies that $y$ is feasible for the unrelaxed problem (9), and so (9) and (10) must have the same optimal value.

Having shown that we may replace the convex program (9) with a linear equivalent (10), we can write this latter problem in the matrix form

$$\max_{y \in \mathbb{R}^n, \ \sigma_1,\dots,\sigma_L \in \mathbb{R}^d} c^\top y \qquad \text{s.t.} \qquad A \begin{bmatrix} y \\ \sigma_1 \\ \vdots \\ \sigma_L \end{bmatrix} \leq b$$

where

$$A = \begin{bmatrix} -I_d & & & \\ & \ddots & & \\ & & -I_d & \\ V_0 & -I_d & & \\ \vdots & W_1 & \ddots & \\ \vdots & & \ddots & -I_d \\ V_L & & & W_L \end{bmatrix} \in \mathbb{R}^{(2Ld+1)\times(n+Ld)}, \qquad b = \begin{bmatrix} \mathbf{0}_d \\ \vdots \\ \mathbf{0}_d \\ -b_0 \\ \vdots \\ -b_{L-1} \\ q - b_L \end{bmatrix} \in \mathbb{R}^{2Ld+1}. \tag{11}$$

By strong duality, if this linear program has an optimal solution, its optimal value is equal to the optimal value of its dual problem:

$$\min_{\nu \in \mathbb{R}^{2Ld+1}} b^\top \nu \qquad \text{s.t.} \qquad A^\top \nu = \begin{bmatrix} c \\ \mathbf{0}_{Ld} \end{bmatrix}, \quad \nu \geq 0. \tag{12}$$

We can incorporate this dual problem (12) into the outer minimization of (1) to yield the non-robust form (8). For a more interpretable form of this dual problem, let $\nu^{(i)}$ denote the portion of the dual vector $\nu$ corresponding to the $i$-th block-row of matrix $A$, indexed from 0. That is, $\nu^{(i)} = \nu_{id+1:(i+1)d}$ for $i = 0, \dots, 2L-1$. Furthermore, let $\mu = \nu_{2Ld+1}$ be the last entry of $\nu$. Written out, the dual problem (12) becomes

$$\min_{\nu^{(0)},\dots,\nu^{(2L-1)} \in \mathbb{R}^d, \ \mu \in \mathbb{R}} \quad \mu(q - b_L) - \sum_{l=0}^{L} b_l^\top \nu^{(L+l)}$$

$$\text{s.t.} \quad \begin{bmatrix} V_0^\top & \cdots & V_L^\top \end{bmatrix} \nu_{Ld+1:} = c$$
$$W_{l+1}^\top \nu^{(L+l+1)} - \nu^{(L+l)} - \nu^{(l)} = \mathbf{0}_d \qquad \forall l = 0, \dots, L-1$$
$$\nu \geq 0.$$

## C.4 Ensuring feasibility of the PICNN maximization problem

As noted at the end of Section 4, it may sometimes be the case that the inner maximization problem of (1) is unbounded or infeasible when $\Omega_\theta(x)$ is parametrized by a PICNN, since in general, the sublevel sets of the PICNN might be unbounded, or the $q$ selected by the split conformal procedure detailed in Section 3.2 may be sufficiently small that $\Omega_\theta(x) = \{\hat{y} \in \mathbb{R}^n \mid s_\theta(x, \hat{y}) \leq q\}$ is empty for certain inputs $x$. We can address each of these concerns using separate techniques.

**Ensuring compact sublevel sets.** To ensure that the PICNN-parametrized score function $s_\theta(x, y)$ has compact sublevel sets in $y$, we can redefine the output layer by setting $V_L = \mathbf{0}_{1 \times n}$ and adding a small $\ell^\infty$ norm term penalizing growth in $y$:

$$s_\theta(x, y) = W_L \sigma_L + \epsilon \|y\|_\infty + b_L, \tag{13}$$

where $\epsilon \geq 0$ is a small penalty term, and where all the remaining parameters and layers remain identical to their definition in (7). This modification ensures that, for any fixed $x$, $s_\theta(x, y)$ has compact sublevel sets, since $\sigma_L \geq 0$ by construction (7) and the penalty term $\epsilon \|y\|_\infty$ will grow unboundedly large as $y$ goes to infinity in any direction, so long as $\epsilon > 0$. Moreover, so long as $\epsilon$ is chosen sufficiently small and the PICNN

is sufficiently deep, this modification should not negatively impact the ability of the PICNN to represent general compact convex uncertainty sets.

Using this modified PICNN (13), the maximization problem (9) can be written as an equivalent linear program

$$
\max_{\substack{y \in \mathbb{R}^n, \ \sigma_1, \dots, \sigma_L \in \mathbb{R}^d, \\ \kappa \in \mathbb{R}}} \quad c^\top y \tag{14a}
$$

$$
\text{s.t.} \quad \sigma_l \geq \mathbf{0}_d \qquad\qquad \forall l = 1, \dots, L \tag{14b}
$$
$$
\sigma_{l+1} \geq W_l \sigma_l + V_l y + b_l \qquad \forall l = 0, \dots, L-1 \tag{14c}
$$
$$
\kappa \geq y_i, \ \kappa \geq -y_i \qquad\qquad \forall i = 1, \dots, n \tag{14d}
$$
$$
W_L \sigma_L + \epsilon \kappa + b_L \leq q \tag{14e}
$$

where the equivalence between (9) and (14) follows the same argument as that employed in the previous section when showing the equivalence of (9) and (10). We can thus likewise apply strong duality to obtain an equivalent minimization form of the problem (14) and incorporate this into the outer minimization of (1) to yield a non-robust problem of the general form (8).

**Ensuring $\Omega_\theta(x)$ is nonempty.** If the $q$ chosen by the split conformal procedure is too small such that $\Omega_\theta(x)$ is empty, i.e., $q \leq q_{\min}$ where

$$
q_{\min} := \min_{\hat{y} \in \mathbb{R}^n} s_\theta(x, \hat{y}), \tag{15}
$$

then we simply increase $q$ to $q_{\min}$ so that $\Omega_\theta(x)$ is guaranteed to be nonempty. That is, for input $x$, we set

$$
q = \max \left( \min_{\hat{y} \in \mathbb{R}^n} s_\theta(x, \hat{y}), \ \text{QUANTILE}(\{s_\theta(x_i, y_i)\}_{(x_i, y_i) \in D_{\text{cal}}}, \ 1 - \alpha) \right).
$$

This preserves the marginal coverage guarantee, as increasing $q$ can only result in a larger uncertainty set $\Omega_\theta(x)$.

In theory, $q_{\min}$ varies as a function of $\theta$, and it is possible to differentiate through the optimization problem (15) using the methods from Agrawal et al. (2019) since the problem is convex and $s_\theta$ is assumed to be differentiable w.r.t. $\theta$ almost everywhere. However, to avoid this added complexity, in practice, we treat $q_{\min}$ as a constant. In other words, on inputs $x$ where we have to increase $q$ to $q_{\min}$, we treat $\frac{\partial q}{\partial \theta} = 0$.

## D   Appendix: Exact differentiable conformal prediction

In this section, we prove how to exactly differentiate through the conformal prediction procedure, unlike the approximate derivative first introduced in Stutz et al. (2022).

**Theorem D.1.** *Let $\alpha \in (0, 1)$ be a risk level, and let $s_i := s_\theta(x_i, y_i)$ denote the scores computed by a score function $s_\theta : \mathbb{R}^m \times \mathbb{R}^n \to \mathbb{R}$ over data points $\{(x_i, y_i)\}_{i=1}^M$. Suppose $s_\theta(x_i, y_i)$ is differentiable w.r.t. $\theta$ for all $i = 1, \dots, M$.*

*Define $s_{M+1} := \infty$. Let $\sigma : \{1, \dots, M+1\} \to \{1, \dots, M+1\}$ denote the permutation that sorts the scores in ascending order, such that $s_{\sigma(i)} \leq s_{\sigma(j)}$ for all $i < j$. For simplicity of notation, we may write $s_{(i)} := s_{\sigma(i)}$.*

*Let $q = \text{QUANTILE}(\{s_i\}_{i=1}^M, \ 1 - \alpha)$ where the QUANTILE function is as defined in Algorithm 1. That is, $q = s_{(k)}$, where $k := \lceil (M+1)(1-\alpha) \rceil \in \{1, \dots, M, M+1\}$. If $s_{(k)}$ is unique, then*

$$
\frac{\mathrm{d}q}{\mathrm{d}\theta} = \begin{cases} \frac{\mathrm{d}}{\mathrm{d}\theta} s_\theta(x_{\sigma(k)}, y_{\sigma(k)}), & \text{if } \alpha \geq \frac{1}{M+1} \\ 0, & \text{otherwise.} \end{cases}
$$

*Proof.* First, when $\alpha \in (0, \frac{1}{M+1})$, we have $k = M+1$, so $q = \infty$ is constant regardless of the choice of $\theta$. Thus, $\frac{\mathrm{d}q}{\mathrm{d}\theta} = 0$.

Now, suppose $\alpha \geq \frac{1}{M+1}$. The QUANTILE function returns the $k$-th largest value of $\{s_i\}_{i=1}^M \cup \{\infty\}$. Since we assume $s_{(k)}$ is unique, we have $\frac{\mathrm{d}q}{\mathrm{d}s_{(i)}} = \mathbf{1}[i = k]$. Finally, we have

$$\frac{\mathrm{d}q}{\mathrm{d}\theta} = \sum_{i=1}^{M} \frac{\mathrm{d}q}{\mathrm{d}s_i} \frac{\mathrm{d}s_i}{\mathrm{d}\theta} = \sum_{i=1}^{M} \frac{\mathrm{d}q}{\mathrm{d}s_{(i)}} \frac{\mathrm{d}s_{(i)}}{\mathrm{d}\theta} = \frac{\mathrm{d}s_{(k)}}{\mathrm{d}\theta} = \frac{\mathrm{d}}{\mathrm{d}\theta} s_\theta(x_{\sigma(k)}, y_{\sigma(k)}).$$

$\square$

The two key assumptions in this theorem are that (1) $s_\theta$ is differentiable w.r.t. $\theta$, and (2) $s_{(k)}$ is unique. When $s_\theta$ is a neural network with a common activation function (e.g., ReLU), (1) holds for inputs $(x, y) \in \mathbb{R}^m \times \mathbb{R}^n$ almost everywhere and $\theta$ almost everywhere. Regarding (2), in practice, just as the gradient of the max function is typically implemented without checking whether its inputs have ties, we do not check whether $s_{(k)}$ is unique.

# E  Appendix: Experiment details

Our experiments were conducted across a variety of machines, including private servers and Amazon AWS EC2 instances, ranging from 12-core to 128-core machines. Our ETO experiments benefited from GPU acceleration across a combination of NVIDIA GeForce GTX 1080 Ti, Titan RTX, T4, and A100 GPUs. Our E2E experiments did not use GPU acceleration, due to the lack of GPU support in the `cvxpylayers` Python package (Agrawal et al., 2019).

In all experiments, we use a batch size of 256 and the Adam optimizer (Kingma & Ba, 2015). Models were trained for up to 100 epochs with early stopping if there was no improvement in validation loss for 10 consecutive epochs.

For box and ellipsoid `ETO` baseline models, we performed a hyperparameter grid search over learning rates $(10^{-4.5}, 10^{-4}, 10^{-3.5}, 10^{-3}, 10^{-2.5}, 10^{-2}, 10^{-1.5})$ and L2 weight decay values $(0, 10^{-4}, 10^{-3}, 10^{-2})$. For PICNN `ETO` models we performed a hyperparameter grid search over learning rates $(10^{-4}, 10^{-3}, 10^{-2})$ and L2 weight decay values $(10^{-4}, 10^{-3}, 10^{-2})$.

## E.1  Uncertainty representation

**Box uncertainty.**  Our box uncertainty model uses a neural network $h_\theta$ with 3 hidden layers of 256 units each and ReLU activations with batch-normalization. The output layer has dimension $2n$, where dimensions $1 : n$ predict the lower bound. Output dimensions $n + 1 : 2n$, after passing through a softplus to ensure positivity, represents the difference between the upper and lower bounds. That is,

$$\begin{bmatrix} h_\theta^{\mathrm{lo}}(x) \\ h_\theta^{\mathrm{hi}}(x) \end{bmatrix} = \begin{bmatrix} h_\theta(x)_{1:n} \\ h_\theta(x)_{1:n} + \mathrm{softplus}(h_\theta(x)_{n+1:2n}) \end{bmatrix}.$$

This architecture ensures that $h_\theta^{\mathrm{hi}}(x) > h_\theta^{\mathrm{lo}}(x)$.

In the two-stage `ETO` baseline, we first train $h_\theta$ to estimate the $\alpha/2$- and $(1 - \alpha/2)$-quantiles, so that $[h_\theta^{\mathrm{lo}}(x), h_\theta^{\mathrm{hi}}(x)]$ represents the centered $(1 - \alpha)$-confidence region. Quantile regression is a common method for generating uncertainty sets for scalar predictions by estimating quantiles of the conditional distribution $\mathcal{P}(y \mid x)$ (Romano et al., 2019). For scalar true label $y$, quantile regression models are commonly trained to minimize *pinball loss* (a.k.a. *quantile loss*) where $\beta$ is the quantile level being estimated:

$$\mathrm{pinball}_\beta(\hat{y}, y) = \begin{cases} \beta \cdot (y - \hat{y}), & \text{if } y > \hat{y} \\ (1 - \beta) \cdot (\hat{y} - y), & \text{if } y \leq \hat{y}. \end{cases}$$

To generalize the pinball loss to our setting of multi-dimensional $y \in \mathbb{R}^n$, we sum the pinball loss across the dimensions of $y$: $\mathrm{pinball}_\beta(\hat{y}, y) = \sum_{i=1}^{n} \mathrm{pinball}_\beta(\hat{y}_i, y_i)$.

Our end-to-end (E2E) box uncertainty models use the same architecture as above, initialized with weights from the trained `ETO` model. We found it helpful to use a weighted combination of the task loss and pinball loss during training of the E2E models to improve training stability. In our experiments, we used a weight of 0.9 on the task loss and 0.1 on the pinball loss. The E2E models used the best L2 weight decay from the `ETO` models, and the learning rate was tuned across $10^{-2}$, $10^{-3}$, and $10^{-4}$.

**Ellipsoidal uncertainty.** Our ellipsoidal uncertainty model uses a neural network $h_\theta$ with 3 hidden layers of 256 units each and ReLU activations with batch-normalization. The output layer has dimension $n + n(n+1)/2$, where dimensions $1 : n$ predict the mean $\mu_\theta(x)$ and the remaining output dimensions are used to construct a lower-triangular Cholesky factor $L_\theta(x)$ of the covariance matrix $\Sigma_\theta(x) = L_\theta(x)L_\theta(x)^\top$. We pass the diagonal entries of $L_\theta(x)$ through a softplus function to ensure strict positivity, which then ensures $\Sigma_\theta(x)$ is positive definite.

For the `ETO` baseline, we trained the model using the negative log-likelihood (NLL) loss

$$\mathrm{NLL}(\theta) = \frac{1}{N} \sum_{(x,y) \in D} -\ln \mathcal{N}(y \mid \mu_\theta(x), \Sigma_\theta(x)),$$

where $\mathcal{N}(\cdot \mid \mu, \Sigma)$ denotes the density of a multivariate normal distribution with mean $\mu$ and covariance matrix $\Sigma$.

Our end-to-end (E2E) ellipsoidal uncertainty models use the same architecture as above, initialized with weights from the the trained `ETO` model. We found it helpful to use a weighted combination of the task loss and NLL loss during training of the E2E models to improve training stability. In our experiments, we used a weight of 0.9 on the task and 0.1 on the NLL loss. The E2E models used the best L2 weight decay from the `ETO` models, and the learning rate was tuned across $10^{-2}$, $10^{-3}$, and $10^{-4}$.

**PICNN uncertainty.** Our PICNN has 2 hidden layers with ReLU activations.

For the battery storage problem, we used 64 units per hidden layer. We did not run into any feasibility issues for the PICNN maximization problem, so we did not restrict $V_L$ as described in Appendix C.4, and we set $\epsilon = 0$.

For the portfolio optimization problem, we tried 32, 64, and 128 units per hidden layer, finding that 32 units worked best. We did run into feasibility issues for the PICNN maximization problem, which we resolved by setting $V_L = \mathbf{0}_{1 \times n}$ as described in Appendix C.4. This change alone was sufficient, and we set $\epsilon = 0$.

For the `ETO` baseline, we take inspiration from the approach by Lin & Ba (2023) to give probabilistic interpretation to a PICNN model $s_\theta$ via the energy-based model $\hat{\mathcal{P}}_\theta(y \mid x) = \frac{1}{Z_\theta(x)} \exp(-s_\theta(x,y))$ where $Z_\theta(x) := \int_{\tilde{y} \in \mathbb{R}^n} \exp(-s_\theta(x,\tilde{y})) \, d\tilde{y}$ is the normalizing constant. We train our `ETO` PICNN models with an approximation to the true NLL loss based on samples from the Metropolis-Adjusted Langevin Algorithm (MALA), a Markov Chain Monte Carlo (MCMC) method. We refer readers to our code for the specific hyperparameters and implementation details we used.

Note that under this energy-based model, adding a scalar constant $c$ to the PICNN (i.e., $s_\theta(x,y) + c$) does not change the probability distribution. That is, $\exp(-s_\theta(x,y)) \propto \exp(-s_\theta(x,y) + c)$. To regularize the PICNN model, which has a bias term in its output layer, we therefore introduce a regularization loss of $w_{\mathrm{zero}} \cdot s_\theta(x,y)^2$ where $w_{\mathrm{zero}}$ is a regularization weight. This regularization loss encourages $s_\theta(x,y)$ to be close to 0, for all examples in the training set. In our experiments, we set $w_{\mathrm{zero}} = 1$.

Our end-to-end (E2E) PICNN uncertainty models use the same architecture as above, initialized with weights from the the trained `ETO` model. Unlike for box and ellipsoidal uncertainty which used a weighted combination of task loss and NLL loss, our E2E PICNN uncertainty models are trained only with the task loss. Similar to the `ETO` PICNN model, we also regularize the E2E PICNN. Here, we add a regularization loss of $w_{\mathrm{q}} \cdot q^2$, where $w_{\mathrm{q}}$ is a regularization weight and $q$ is the conformal prediction threshold computed in each minibatch of E2E training. This regularization loss term aims to keep $q$ near 0; without this regularization, we found that $q$ tended to grow dramatically over training epochs with poor task loss. In our experiments, we set $w_{\mathrm{q}} = 0.01$.

The E2E models used the best L2 weight decay from the `ETO` models. For the battery storage problem, we tested learning rates of $10^{-3}$ and $10^{-4}$. For the portfolio optimization problem, we used a learning rate of $5 \times 10^{-3}$.

## E.2 Data

**Price forecasting for battery storage.** We use the same dataset as Donti et al. (2017) in our price forecasting for battery storage problem. In this dataset, the target $y \in \mathbb{R}^{24}$ is the hourly PJM day-ahead system energy price for 2011-2016, for a total of 2189 days. Unlike Donti et al. (2017), though, we do not exclude any days whose electricity prices are too high ($>500\$/MWh$). Whereas Donti et al. (2017) treated these days as outliers, our conditional robust optimization problem is designed to output robust decisions. For predicting target for a given day, the inputs $x \in \mathbb{R}^{101}$ include the previous day's log-prices, the given day's hourly load forecast, the previous day's hourly temperature, the given day's hourly temperature, and several calendar-based features such as whether the given day is a weekend or a US holiday.

For the setting without distribution shift, we take a random 20% subset of the dataset as the test set; because the test set is selected randomly, it is considered exchangeable with the rest of the dataset. For the setting with distribution shift, we take the chronologically last 20% of the dataset as the test set; because load, electricity prices, and temperature all have distribution shifts over time, the test set is not exchangeable with the rest of the dataset. Figure A4 illustrates the data from the distribution shift setting, where the electricity price tends to be lower and has lower variability in the test set compared to the training/validation splits. For each seed, we further use a 80/20 random split of the remaining data for training and calibration.

**Portfolio optimization.** For the portfolio optimization task, we used synthetically generated data. We sample $x \in \mathbb{R}^2, y \in \mathbb{R}^2$ from a mixture of three 4-D multivariate Gaussian distributions as used in Chenreddy & Delage (2024). Formally,

$$\begin{bmatrix} x \\ y \end{bmatrix} \sim p_a \, \mathcal{N}(\mu_a, \Sigma_a) + p_b \, \mathcal{N}(\mu_b, \Sigma_b) + p_c \, \mathcal{N}(\mu_c, \Sigma_c)$$

where $p_a + p_b + p_c = 1$. Specifically,

$$p_a = \phi, \qquad p_b = \frac{1}{\alpha_{\mathrm{GMM}} + 1}(1 - \phi), \qquad p_c = \frac{\alpha_{\mathrm{GMM}}}{\alpha_{\mathrm{GMM}} + 1}(1 - \phi),$$

$$\mu_a = \mathbf{0}_4, \qquad \mu_b = \begin{bmatrix} 0 & 5 & 5 & 0 \end{bmatrix}^\top, \qquad \mu_c = \mu_b,$$

$$\Sigma_a = \begin{bmatrix} 1 & 0 & 0.37 & 0 \\ 0 & 1.5 & 0 & 0 \\ 0.37 & 0 & 2 & 0.73 \\ 0 & 0 & 0.73 & 3 \end{bmatrix}, \qquad \Sigma_b = \alpha_{\mathrm{GMM}} \Sigma_a, \qquad \Sigma_c = \frac{1}{\alpha_{\mathrm{GMM}}} \Sigma_a,$$

for some $\phi \in [0, 1]$ and $\alpha_{\mathrm{GMM}} \in [0, 1]$. In our experiments, we used $\phi = 0.7$ and $\alpha_{\mathrm{GMM}} = 0.9$. (Chenreddy & Delage (2024) do not disclose the values of $\phi$ and $\alpha_{\mathrm{GMM}}$ chosen for their experiments.) For each random seed, we generate 2000 samples and use a (train, calibration, test) split of (600, 400, 1000).

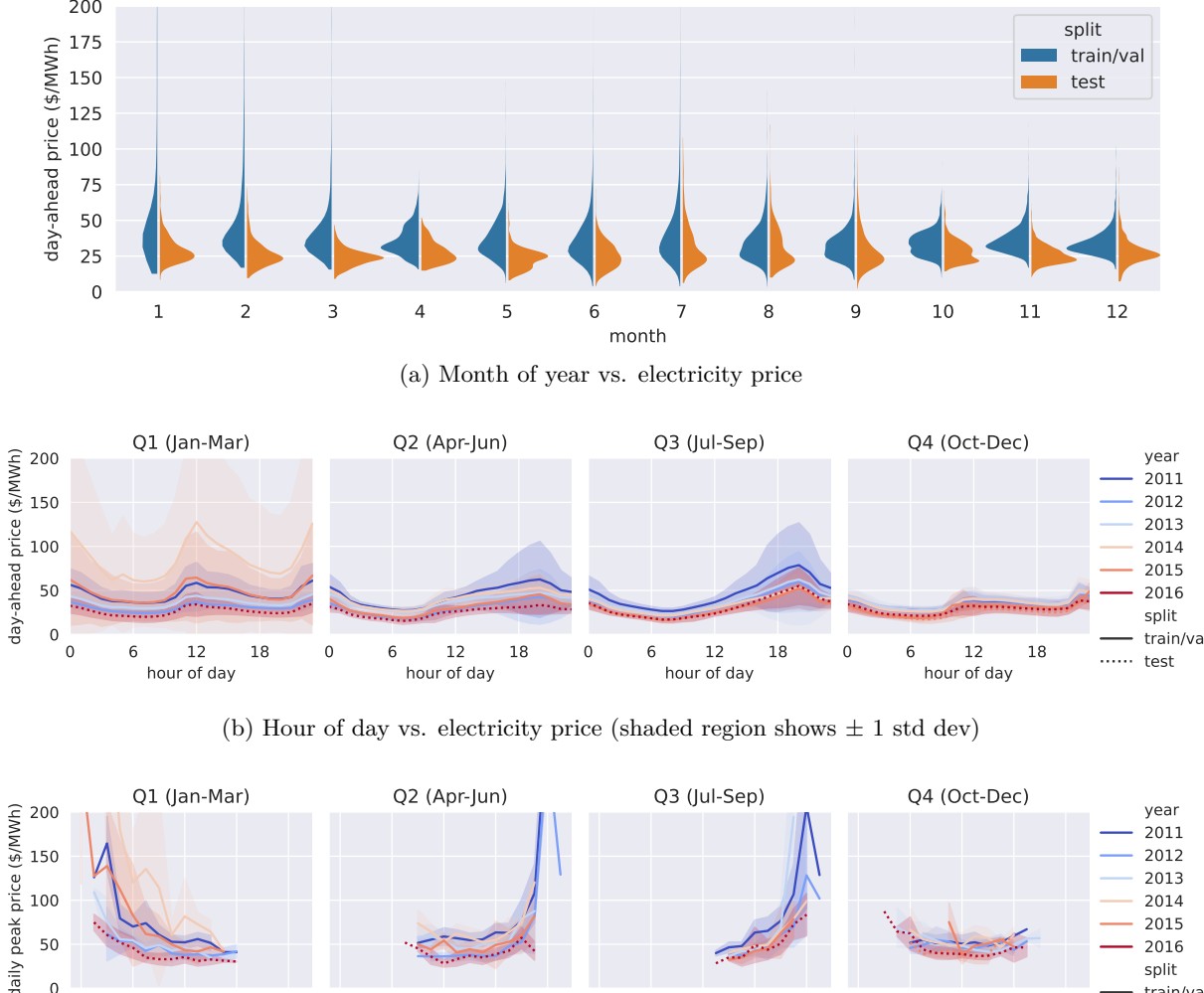

(a) Month of year vs. electricity price

(b) Hour of day vs. electricity price (shaded region shows $\pm$ 1 std dev)

(c) Daily high temperature (rounded to nearest $5^\circ$F) vs. daily peak electricity price (shaded region shows $\pm$ 1 std dev

Figure A4: Visualization of data from the battery storage problem under distribution shift. The train/val splits are sampled randomly from the range 2011-01-04 to 2015-10-20, whereas the test set comprises 2015-10-21 to 2016-12-31. These plots evidently show that the electricity price is generally lower in the test set and also has lower variability by both hour of day and the daily maximum temperature.

