# OpenReview forum: "End-to-End Conformal Calibration for Optimization Under Uncertainty"
_TMLR — Accepted by TMLR_

### Review · Reviewer_qLpG · 2025-07-18

**Summary Of Contributions:**

The authors propose a novel methodology for learning decision-aware uncertainty representations for conditional robust optimization. The primary contributions are threefold:

* The paper develops a novel end-to-end (E2E) training framework that directly accounts for the downstream task loss. By backpropagating through both the robust optimization problem and a conformal calibration procedure, the framework learns uncertainty sets that are explicitly shaped by their utility in the decision-making context. This presents a significant departure from conventional estimate-then-optimize (ETO) paradigms, which decouple the prediction and optimization stages.

* The authors propose the use of Partially Input-Convex Neural Networks (PICNNs) to parameterize general convex uncertainty sets. This approach enhances the expressiveness of the uncertainty representation beyond standard approaches e.g., boxes or ellipsoids.

* The work introduces an efficient method for differentiating through the quantile function. This technique avoids the need for smoothed approximations used in prior work, enabling precise gradient-based learning while preserving the finite-sample coverage guarantees of conformal prediction.

**Audience:**

Yes

**Broader Impact Concerns:**

I have no concerns regarding the ethical implications of the work.

**Claims And Evidence:**

Yes

**Requested Changes:**

* **Dataset description**: I am not familiar with the specific datasets used by authors. However, they mention that the price forecast for the battery storage problem is conditioned on the price on the past day. I wonder whether that violates the exchangeability assumption given that the dataset seems to comprise time series.

* **Deepen the Discussion on Distribution Shift**: In Section 5.4, the authors may expand the analysis of why the more expressive models exhibit worse coverage under distribution shift. A plausible hypothesis is that their flexibility allows them to overfit to spurious correlations in the training distribution that do not generalize. Suggesting potential avenues for mitigation (e.g., specific regularization techniques, domain adaptation methods) would strengthen this section.

* **Reproducibility**: authors may consider to move crucial details about the hybrid loss functions and regularization from the appendix into the main methodology (Section 3.3 or 4). That would provide readers with a better understanding of what is required to successfully train these models and strengthens the paper's contribution to practical application.

* **Computational Requirements**: (related to the previous point) - the authors may add more details concerning scalability limitation of their approach. E.g., concerning the optimization library used. This is an important practical consideration for other researchers looking to build on this work.

**Strengths And Weaknesses:**

**Strengths**

* **Novelty and Significance**: The work presents an interesting approach to bridge the gap between uncertainty quantification and downstream optimization tasks. The proposed end-to-end framework, which directly incorporates the task loss into the learning of uncertainty sets, offers a compelling alternative to the prevailing estimate-then-optimize (ETO) paradigm.

* **Theoretical Guarantees**: The framework is theoretically sounding. It provides formal, finite-sample marginal coverage guarantees for its uncertainty sets by leveraging split conformal prediction.

* **Methodological Soundness**: The proposed use of Partially Input-Convex Neural Networks (PICNNs) to parameterize general convex uncertainty sets is methodologically sound.

* **Rigorous Empirical Validation**: The claims are substantiated by empirical validation on two relevant applications: energy storage arbitrage and portfolio optimization. The framework's performance is superior to multiple ETO baselines across various uncertainty models and risk levels.


**Weaknesses and Areas for Improvement**

* **Performance Degradation Under Distribution Shift**: The empirical results show the framework's performance degradation under distribution shift, where the exchangeability assumption essential for conformal prediction is violated. The observation that more expressive models exhibit a more significant drop in coverage is important. However, the analysis could be deepened by investigating the mechanisms behind this vulnerability rather than primarily noting it as an avenue for future work

* **Practical Complexity and Training Stability**: From the appendix, it seems that the practical implementation of the end-to-end training procedure is non-trivial and sensitive to the optimization setup. For example, as detailed in the appendix, stable training necessitated the use of specific regularization of the conformal quantile for the PICNN model to prevent divergence. This suggests that the decision-focused loss landscape is challenging to optimize directly and that the reported performance may depend on these carefully chosen heuristics.

---

> ### Author Response · Authors · 2025-08-28
> **Response to Reviewer qLpG**
>
> We thank the reviewer for thoughtful and constructive feedback. We respond to each of the reviewer's points below.
>
> **Distribution shift:** We would like to emphasize that analyzing distribution shifts and correcting for such shifts is not the main point of our work. We present results on the distribution shift setting for two reasons:
> 1. We want to highlight that in general, conformal prediction coverage guarantees do not hold under distribution shift, as is common with time series data.
> 2. Even under distribution shift, our E2E approach performs better than the ETO baselines. This suggests that the performance gains of the E2E approach do not come at the cost of overfitting the data distribution.
>
> Many existing methods such as likelihood weight (Tibshirani et al., 2019) have been developed to account for distribution shifts when applying conformal prediction methods. We believe that studying how these methods can be incorporated into the conditional robust optimization problem is a useful avenue for future research, but this is not the focus of our present work.
>
> **Practical complexity, training stability, and reproducibility:** The reviewer correctly points out that the decision-focused loss landscape is sometimes challenging to optimize directly. Indeed, most published papers in the decision-focused learning (DFL) or predict-then-optimize literature either use a weighted combination of a decision-focused loss and a standard loss function, or they only use DFL for fine-tuning a model that has been pre-trained with a standard loss. As we describe in the appendix, we follow both of these standard heuristics. We have added a paragraph at the end of Section 3.3 to highlight the heuristics.
>
> **Dataset description:** In general, the reviewer is correct that time series data is generally not exchangeable. This is indeed the case for our experiments that we label, "with distribution shift." For our experiments labeled "no distribution shift," the entire dataset is randomly shuffled before splitting into train, calibration, and test splits. The random data shuffling, by construction, produces an exchangeable dataset.
>
> **Computational requirements:** Following the reviewer's suggestion, we have added a detailed description of computational performance in Appendix A.1, including the packages and solvers used for our conditional robust optimization prolems.

---

### Review · Reviewer_iVud · 2025-07-18

**Summary Of Contributions:**

This paper focuses on optimizing uncertainty-aware decision making in a more holistic, trainable fashion. The paper combines conformal prediction (to generate uncertainty sets) with a decision-based task loss to obtain a min-max optimization objective that incorporates conformal and task information, and obtains subsequent loss gradients to update the underlying model. The paper makes convexity assumptions on the objectives and employs a partial-convex neural network (PICNN) to render the optimization tractable. The PICNN models the conformal scoring function in a flexible, convex fashion, permitting generating multi-variate convex uncertainty sets that improve upon simpler box or ellipsoidal sets. A synthetic 2D portfolio experiment is used to visually motivate improvements over two baselines, and tabular regression on energy pricing time series (re-shuffled to ensure exchangeability) further motivates improvements on the task loss (suggesting better decision-making) while keeping conformal coverage.

**Audience:**

Yes

**Broader Impact Concerns:**

I have no ethical concerns regarding the paper.

**Claims And Evidence:**

Yes

**Requested Changes:**

- The proof of Prop. 3.3 is rather a comment or description highlighting where the results come from, but not a proper proof. A more explicit proof should be included in the Appendix for completeness. Also, more details on the conformal training approach taken from Stutz et al. should be added to the Appendix, as this forms an important part of the pipeline.
- Visualizations of the obtained conformal uncertainty sets/intervals for the battery experiment would be good to have.

Other comments
- Nonconformity score uniqueness could also be ensured with some extra jitter in practice (instead of simply trusting in it).
- What is the intuition behind permitting negative task losses?
- Is my understanding correct that the task loss reduces (Fig. 3) with higher uncertainty level alpha because the optimization becomes less restrictive, i.e. the effect of the inner max. is lessened?

**Strengths And Weaknesses:**

Strengths
- The paper is quite well written and walks the reader coherently through the problem statement and proposed solution, highlighting the arising research questions and proposals to address them. I also appreciate the mix of visualizations and theoretical backing, and the proper explanations and related work review in the appendix.
- The problem of rendering these models more valuable for down-stream decision-making is important. Many papers end with the conformal procedure and motivate obtained sets as useful for decision-making, but do not go the extra steps of actually modeling decision-making and addressing this aspect. So this direction is good for furthering the field.

Weaknesses
- The paper does not talk much about the weaknesses or limitations of the approach, both in terms of imposed convexity restrictions (Ass. 3.1), and the limitations of proposing an end-to-end framework restricted to the particular input-convex model (PICNN). In essence, this approach loses most of the fundamental appeals of conformal prediction, i.e. it’s post-hoc, modelling-free, training-free nature. The appeals of the particular modelling-heavy pipeline should be motivated further (especially in light of limited experimentation), or the scope properly set to the types of problems where this does not constitute any major limitations. I’d be happy to learn more about the range of use cases the authors have in mind.
- Experimental validation, while claimed to “extensively evaluate” and “demonstrate conclusively” only contains one synthetic experiment and one tabular regression task, with two external baselines (ETO-SLL, ETO-JC). Furthermore, baseline comparisons are limited to particular settings, e.g. PICNN only compares internal baselines, and the only full comparison is on ellipsoidal sets. Two different groups of baselines seem relevant: 1) when it comes to multivariate prediction sets, the literature offers a substantially wider range of shapes beyond box/ellipse, e.g. consider recent comparison (Dheur et al. 2025). This includes similarly flexible but non-convex approaches; 2) as internalization of CP into the training objective is performed, it seems that a fair comparison should similarly incorporate conformal training (Stutz et al. ) or related uncertainty “tuning” methods like conformal boosting (Ran et al. 2024) across baselines. When it comes to performed experiments, while interesting they are mainly limited to the iid data case, and I find the shift results not very convincing especially in light of employing actual time series data. More experiments highlighting realistic use cases should be done to help highlight the usefulness of going the modelling-specific route. For example, in Tab. 1 all results are well within standard deviations and obtained task loss benefits (and visualized uncertainty sets) seem questionable. Overall, more experiments are needed and the claims on empirical validation/advantage should be softened.

Method Clarifications / Questions
- Perhaps I am misunderstanding but does the PICNN include both prediction and actually modeling the score function? Is any distinction in that matter made? When we update the model parameters, should I interpret this as underlying “finetuning” or rather updates to the conformal calibration procedure only? I was confused about this looking at Fig. 1.
- ETO is invalidated as lacking task loss feedback. But instead of doing gradient-based model updates, could we not simply feed task loss information into the conformal calibration procedure by designing smart nonconformity scores accounting for task loss, or doing some other form of post-hoc quantile adjustments?
- Looking at Algo. 1, do I understand correctly that a conformal quantile and solution of the optimization problem needs to be recomputed for every test input at inference time? If so, any comments on computational costs? And if so, why do we separate a train and inference step but not simply continuously update the model parameters by computing gradients (as we already solve the optimization) to adapt the model even further, potentially robustifying to slow shifts?
- In sec. 3.3., is there a reason to do the end-to-end training with ERM? Can we not directly go for some more robust optimization procedure here? I perhaps misunderstood how this relates to the “robust” optimization objective Eq. 1.
- Fig. 4 displays marginal coverage levels and states “confirming that the improvements in task loss performance from our E2E approach do not come at the
cost of worse coverage” . From my understanding of the method after the training/updating step, split CP is simply added on top during inference, so the coverage rate by the conformal quantile does not directly relate to the task loss and is trivially satisfied (when exchangeable). Am I missing the connection to the task here?

References
- Dheur, Victor, et al. "Multi-Output Conformal Regression: A Unified Comparative Study with New Conformity Scores." arXiv preprint arXiv:2501.10533 (2025).
- Xie, Ran, Rina Barber, and Emmanuel Candes. "Boosted conformal prediction intervals." Advances in Neural Information Processing Systems 37 (2024): 71868-71899.

---

> ### Author Response · Authors · 2025-08-28
> **Response to Reviewer iVud, Part 1/3**
>
> We thank the reviewer for thoughtful and constructive feedback. We respond to each of the reviewer's points below.
>
> **Convexity Limitation**: The reviewer correctly points out that convexity is a key assumption in our approach. Convexity shows up in two places:
>
> 1. The uncertainty set $\Omega(x)$ must be convex in order for the inner maximization optimization problem to be tractable. This is a standard assumption in the robust optimization literature. Furthermore, we would like to highlight that our contribution of using a sublevel set of a partial input-convex neural network (PICNN) as the representation of a conditional/contextual uncertainty set was meant to address limitations in traditional approaches which rely on even more restrictive box and ellipsoidal uncertainty sets.
> 2. The outer problem must be convex in $z$. The specific form of $f$ described in Assumption 3.1 is convenient to derive an exact dual of the inner problem; technically, our approach can be easily extended to support any conic-representable saddle function $f$. We have updated the manuscript with a note about this point.
>
> **Losing appeal of conformal prediction**: We agree with the reviewer that the split conformal prediction procedure is appealing due to two key properties:
>
> 1. Split conformal prediction is "distribution-free," meaning that it makes no assumption on the underlying joint distribution over $(x,y)$. Our approach directly inherits this property as well.
> 2. Split conformal predition is applied post-hoc, thereby requiring no training. In some settings, this is a benefit. However, we show that this is actually a limitation of naively applying split conformal prediction, because it leaves a significant amount of performance on the table. Our experiments demonstrate that training end-to-end by differentiating through the split conformal procedure provides significant performance gains. Of course, whether the performance gain is worth the added computation cost is application-dependent.
>
> **Experimental validation**
> 1. We have replaced "extensively evaluate" with "evaluate" and "demonstrate conclusively" with "demonstrate" to tone down our claims.
> 2. As noted in our related works section, the two external baselines ETO-SLL and ETO-JC are the primary estimate-then-optimize approaches that have guarantees. The reason PICNN only compares to internal baselines is because we are the first work to propose using a PICNN-based uncertainty set representation. No prior methods used PICNN representations; the ETO-SLL/ETO-JC approaches are specific to box and/or ellipsoidal sets.
> 3. We appreciate the reviewer pointing us to the recent comparison by Dheur et al. (2025) of conformal multivariate prediction sets, which can be categorized as either hyperrectangles, density superlevel sets, union of balls, or quantile regions. Of these sets, the density superlevel sets and quantile regions are generally non-convex, which do not generally yield tractable robust optimization problems. The "union of balls" parameterization, while also nonconvex, may be used for robust optimization, as demonstrated by Patel et al. (2024), which we mentioned in our Related Works section. However, the complexity of the robust optimization problem grows linearly in the number of balls; as we show in our new performance analysis (see Table A1 in the Appendix), the main computational cost in ETO and E2E comes from solving the robust optimization problem, so using many balls to represent complicated sets may be prohibitively expensive in computational cost. Finally, the hyperrectangle sets from M-CP and CopulaCPTS are similar to our "box" uncertainty sets.

---

> ### Author Response · Authors · 2025-08-28
> **Response to Reviewer iVud, Part 2/3**
>
> **Experimental validation, continued**
>
> 4. The reviewer suggests comparing against the "conformal training" method by Stutz et al. (2022), but a direct comparison is infeasible because the conformal training method was developed for classifiers, whereas our method is for regression problems. Furthermore, we would like to make several additional remarks about conformal training.
>    - Broadly speaking, the conformal training procedure can be thought of training models end-to-end to minimize the size of prediction sets. We would like to point out, though, that the size of the prediction set is not necessarily a useful loss function. As we illustrate in Figure A3, smaller prediction sets do not necessarily result in lower robust decision costs.
>    - Our work is inspired by the conformal training procedure. We adapt the pseudo-split-conformal procedure during each minibatch of training for our E2E algorithm. The key difference is that we use the decision cost as the loss function, instead of the size of the prediction set.
>    - We would also like to highlight that while it may be possible to minimize the volume or diameter of prediction sets for simple shapes (e.g., box or ellipsoids), it is generally intractable to compute the volume or diameter of an arbitrary polytope, such as a sublevel set of a PICNN.
> 5. The reviewer also suggests comparing against other methods that "tune" the nonconformity score function, such as conformal boosting (Xie et al. 2024). While this is an interesting suggestion, we have not implemented this comparison for the four reasons specified below. If the reviewer would like, we may include the following discussion in our revision.
>    - First, the conformal boosting paper primarily considers the objectives of reducing prediction set size and/or improving conditional coverage for univariate labels. As we discussed in point (4) above, we have shown that smaller prediction sets do not necessarily result in lower decision costs. It is also not necessarily true that improved conditional converage provides lower decision costs.
>    - Second, even if try to generalize conformal boosting to multivariate labels and repurpose conformal boosting for "learning" or "tuning" the nonconformity score function to align with robust optimization, the conformal boosting approach is likely at least as computationally intensive during training as our E2E training approach, if not more intensive. Each round of boosting requires resolving the robust optimization problem on all of the training and validation data points. As shown in Table A1, the time spent solving the robust optimization problems dominates the E2E computational cost. Thus, if the number of boosting rounds and the number of E2E training epochs is the same, then their computation is roughly the same. If one chooses to further apply the $k$-fold cross-validation procedure suggested in the conformal boosting paper, then the computational cost of conformal boosting becomes $k$ times more costly than our E2E approach.
>    - Third, our E2E approach keeps the same model architecture as the two-stage ETO models, whereas conformal boosting adds additional post-hoc processing boosted "layers", where the # of new "layers" is the # of boosting rounds. Thus, our E2E approach yields a model that is faster at inference time compared to conformal boosting.
>    - Fourth, boosting is generally limited to a simple hypothesis family. Each score function in the family proposed by Xie et al. only results in convex box-shaped uncertainty sets; while the sum of many of these functions may be able to approximate more complex convex regions, doing so would require many rounds of boosting. In contrast, our approach _directly_ learns a general convex uncertainty set, leveraging the fact that ICNNs are proven universal convex function approximators.
> 6. Regarding the distribution shift experiments involving time series data, we point the reviewer to our response to reviewer A4ot. In short, in our updated revision we now plot the time series data to highlight the kind of distribution shift that is present.
> 7. Regarding the portfolio optimization problem, while the best E2E results are sometimes within the reported standard deviation of the best ETO results, they are generally not within the standard error (which divides the stddev by $\sqrt{n}$ where $n=10$). If the reviewer would like, we can report standard error instead of standard deviation. Furthermore, there is nonetheless a clear trend of E2E being better than ETO across *all* uncertainty levels $\alpha$. Moreover, our results on the battery storage problem clearly exhibit significant improvement of E2E over ETO which in most cases falls well outside the standard deviation interval.

---

> ### Author Response · Authors · 2025-08-28
> **Response to Reviewer iVud, Part 3/3**
>
> **Method Clarification**
>
> 1. The PICNN directly models the nonconformity score function and eschews any point estimate; the PICNN _is_ the nonconformity score function. Our E2E PICNN procedure uses gradient descent to try to learn the best nonconformity score function for minimizing the decision loss. Thus, the E2E PICNN procedure is both "finetuning" the PICNN model _and_ updating the conformal calibration score function.
> 2. Indeed it may be possible to design nonconformity score functions to account for the conditional robust optimization task loss, without performing gradient-based model updates. However, to the best of our knowledge, there have not been published works demonstrating this in the regression setting. We have added a paragraph in our related works section to highlight that Kiyani et al. (2025) and Cortes-Gomez et al. (2024) have designed task-loss based nonconformity score functions for classification problems. However, the key challenge in regression is designing a flexible/expressive nonconformity score function while ensuring that the resulting uncertainty set is convex for tractable use in robust optimization. A PICNN meets both of these criteria, and fitting this PICNN then naturally suggests using a gradient-based approach; indeed, this is how we developed our E2E PICNN idea.
> 3. The inference procedures (and therefore computational costs) for the ETO and E2E approaches are identical. At inference time, we compute a conformal quantile $q$ via a forward pass of the model on the calibration set. Typically, a calibration set size of 100-1000 points is sufficient, and thus the computation is fast. We only perform this computation _once_, and we use the same $q$ across all test set points. For each test set point, we solve its corresponding conditional robust optimization (CRO) problem. In our examples, solving the CRO problem takes roughly 0.001-0.050 seconds per test point, depending on the parameterization of the uncertainty set. Table A1 in our revision gives some performance metrics.
> 4. We use the expected task loss as our overall training objective to demonstrate how our method can be used to trade-off between the goals of average-case performance and robustness. Our method can also be easily adapted to minimize other risk measures such as the $\delta$-conditional value-at-risk (CVaR). In the case of $\delta$-CVaR, the only modification to our algorithm is that we would only backpropagate gradients through the $(1-\delta)$-fraction of examples per minibatch with the highest task loss. (ERM then corresponds to the case where $\delta = 0$.) We can add a mention of this to the paper if the reviewer would like.
> 5. The reviewer is correct that we add split CP on top during inference, so the marginal coverage rate is guaranteed when the data are exchangeable. Our statement that "lower task loss of our E2E models does not come at the expense of worse coverage" aims to convey two points. (a) In the exchangeable case, our coverage plots are a sanity check that we are indeed seeing the marignal coverage guarantee from split CP for both ETO and E2E. (b) In the distribution shift case, where there is no longer a coverage guarantee, the lower task loss from E2E is not an artifact of lower coverage.
>
> **Requested Changes**
>
> 1. We have updated the proof for Proposition 3.3 to use more formal language.
> 2. Could the reviewer please clarify what additional details on the conformal training approach (from Stutz et al.) the reviewer believes is missing from our paper? We would be happy to include such details in the Appendix.
> 3. While we agree that it would be nice to include visualizations of the conformalized prediction sets from the battery storage experiment, it is unclear to us how to visualize 24-dimensional prediction sets. We welcome suggestions from the reviewer. The naive strategy of treating each hour of the day independently would work for the box prediction sets, but not for the ellipse and PICNN prediction sets. In the ellipse and PICNN sets, the predictions for each hour of the day are dependent, and it is unclear how we may plot that dependence in a meaningful way.
>
> **Other comments**
>
> 1. While some auxiliary random noise can be added to ensure uniqueness of nonconformity scores, we do not believe that to be a meaningful to do, as it would result in non-deterministic prediction sets.
> 2. Our objective, whether in the battery storage problem (technically just the 1st term in the objective) or portfolio optimization, is to maximize profit. However, because it is more common in the machine learning literature to use the terminology of _minimizing loss_, we define our "task loss" as "negative profit." Hence, the task loss is negative when the profit is positive.
> 3. Correct. Larger $\alpha$ generally results in lower average task loss because the optimization is less conservative.

---

### Review · Reviewer_A4ot · 2025-07-25

**Summary Of Contributions:**

This paper proposes an end-to-end approach for condition robust optimization. The authors propose an approach based on minimizing the decision-loss while differentiating through conformal prediction to ensure a desired marginal coverage is met.  In addition, the authors propose partially input-convex neural networks (PCINNs) as a means to parameterize uncertainty sets.  This approach is then evaluated on a price forecasting problem for battery storage and a portfolio optimization, where their approach, along with the uncertainty sets learning through PCINNs, achieves both a good task loss and coverage.  In addition, they evaluate their approach within the context of distribution shifts and demonstrate favorable results.

**Audience:**

Yes

**Claims And Evidence:**

Yes

**Requested Changes:**

**W1/2** would strengthen the contribution and provide a more comprehensive evaluation that helps me lean toward acceptance.  **W4/5** issues need to be addressed, but they are minor.  Address.  For **W3**, the authors should either tone down their claims, unless a more robust evaluation is done.

**Strengths And Weaknesses:**

### Strengths
- **[S1]**: Novelty and methodological contributions.  Overall, this paper makes two major contributions.  The first is the integration of differentiating through the conformal prediction within an end-to-end learning context.  The second is the learned uncertainty sets within the context of conditional robust optimization.  These two contributions are both well motivated and a major strength of the paper.
- **[S2]**: Experimental results.  Overall, the numerical results of the paper are promising, although I'll discuss some limitations and suggestions for improvement in the weaknesses.  That said, the end-to-end approach consistently achieves not only good task loss, but also good coverage.  In addition, the authors present some results on distributional shifts that are favorable for the proposed methods.
- **[S3]**:  Clarity and presentation. The paper is well-written.  Contributions are clearly stated, theory is appropriate, and numerical results are presented cleanly.

### Weaknesses
- **[W1]**: Evaluation metrics.  Given the focus on conditional robust optimization (CRO), standard CRO metrics should be reported, such as VaR/CVaR, rather than just the task loss, as these metrics prioritize robustness, i.e., what decision makers are often concerned with.
- **[W2]**: Baselines.  While the authors evaluate and compare their approach with existing ETO approaches, they do not compare it with the other end-to-end approach presented in [1].  The authors state that they only compare against approaches that maintain guaranteed marginal coverage.  However, even the proposed approach only guarantees marginal coverage for the calibration set and not the test set.  For this reason, I believe it would be reasonable to compare to [1], especially given the marginal coverage %'s reported are close to the target marginal coverage.
- **[W3]**: Distribution shifts experiments.  These experiments are not particularly convincing.  The authors split the data temporally.  However, there is no discussion of how the distribution of data differs temporally.  Insight into this, or perhaps providing an ablation on varying levels of shifts through perturbation, would be more convincing.
- **[W4]**: No reporting on times to give insight into scalability.  Given the context of optimization, it would be helpful to report the time required for training between the different variants of methods, e.g., if box/ellipsoid are more efficient in training, they may be preferable in specific contexts.
- **[W5]**:  Assumption 3.1, in particular $g$ not being a function of $y$.  While most existing approaches make the same assumption, this should be stated earlier and as a separate assumption from the assumption on the objective, given that it may easily be missed.  Alternatively, framing the approach as focusing on objective uncertainty and dropping $y$ from the constraints would make the notation more consistent with the approach's limitations.

---

> ### Author Response · Authors · 2025-08-28
> **Response to Reviewer A4ot**
>
> We thank the reviewer for thoughtful and constructive feedback. We respond to each of the reviewer's points below.
>
> **[W1]:** In our revision, we provide plots of the $(1-\alpha$)-VaR and $(1-\alpha)$-CVaR of the task loss obtained via our method on the battery storage problem. In particular, for each target coverage level $1 - \alpha$, we evaluate $\mathrm{VaR}^{1-\alpha}[\text{task loss}]$ and $\mathrm{CVaR}^{1-\alpha}[\text{task loss}]$ for both the ETO baseline(s) and our E2E methodologies across box, ellipsoid, and PICNN uncertainty sets. In general, it is clear that our E2E methodology uniformly improves over ETO, and while the performance of E2E ellipsoid is close to that of E2E PICNN, the latter appears to perform better for certain $\alpha$. These results highlight that our methodology improves not only the average task loss, but robustness more generally, even when compared with the robust ETO baselines.
>
> **[W2]:** We are not sure what the reviewer means by [1] and welcome clarification from the reviewer. We also would like to point out that our proposed approach does indeed guarantee marginal coverage on the test set, as long as the test set is exchangeable with the calibration set (for example, if the test set is i.i.d. with the calibration set). This is a standard assumption in the conformal prediction literature.
>
> **[W3]:** In our revision, we have added a figure to the appendix which plots the relevant time series data and clarifies the extent and type of distribution shift present in these experiments. Also, we would like to clarify that the primary purpose of our distribution shift experiments is not to exhaustively assess the performance of our approach under various kinds of distribution shift, but rather to supplement our results in Section 5.3 by highlighting that our end-to-end training methodology's improvement of task loss does not necessarily come at the expense of robustness in real-world settings such as the battery storage problem, when compared to other baseline approaches.
>
> **[W4]:** Following the reviewer's suggestion, we now report the wall-clock time of the different methods in a table in the appendix. For convenience, we reproduce the table below. The "pretrain" column gives the time per epoch of training (in seconds) with standard loss function (pinball loss for Box, negative log-likelihood loss for Ellipse and PICNN). The "optimize" column gives the time needed to compute the decision $z^\star_\theta(x)$ given a pretrained model. The "E2E" column gives the time per epoch of E2E training, which requires computing $z^\star_\theta(x)$ for each training and calibration example. Evidently, the bulk of the time spent on E2E training comes from computing the decision $z_\theta^\star(x)$; the additional overhead from computing the gradient through the KKT conditions of the optimization problem is much less than solving the optimization problem itself.
>
> | | Pre-train | Optimize (train+calib) | E2E |
> |-|-|-|-|
> | Box | 0.03 ± 0.01 | 4.29 ± 0.07 | 6.34 ± 0.32 |
> | Ellipse | 0.03 ± 0.01 | 7.79 ± 0.06 | 7.12 ± 1.28 |
> | PICNN | 4.20 ± 0.20 | 50.49 ± 0.19 | 33.24 ± 0.25 |
>
> We note that in theory, E2E times should always be higher than the optimization time. The optimization time accounts for the time required to compute $z_\theta^\star(x)$, and E2E training additional accounts for the time required to compute the gradient $\frac{d}{d\theta} z_\theta^\star(x)$. However, in practice, the optimization time depends heavily on the choice of numerical solver. In our implementation, we use the default `cvxpy` solver (Clarabel) for the optimization step in ETO, whereas we use the default `cvxpylayers` solver (SCS) during E2E training. In the case of the Ellipse and PICNN uncertainty sets, the SCS solver is generally faster than Clarabel; hence, the following table counterintuitively reports lower E2E training times for Ellipse and PICNN than optimization times.
>
> We further note that the optimization and E2E training times reflect our particular implementation, but significantly faster times are likely possible. In particular, we solve all of the optimization problems (in both the ETO optimization step and during E2E training) using CPU-based solvers; however, recently developed GPU-accelerated solvers may be able to reduce the optimization time required by orders of magnitude, especially when solving a batch of problems with the same structure.
>
> **[W5]:** We agree with the reviewer that it may be confusing to introduce the constraint $g(x,y,z) \leq 0$ in the definition of the conditional robust optimization (CRO) problem, only to remove the dependence on $y$ later on. We have updated the definition of the CRO problem so that the constraint $g$ is only a function of $x,z$. We have also updated Assumption 3.1 and Figure 1 accordingly.

---

> > ### Comment · Reviewer_A4ot · 2025-09-08
> > **Response to authors**
> >
> > My apologies, the missing reference for [1] is provided below.  I'll reiterate the main point of the weakness: no other E2E methods are compared against.  As the authors point out, this approach does not guarantee the coverage requirement, as per their reason for not comparing against [1].  However, empirically, [1] demonstrates that they are typically very close to meeting the coverage requirements, so I believe a comparison would be reasonable.  If the authors include these experiments, it would significantly improve my confidence in the claimed advantages of their proposed method.
> >
> > - [1] Abhilash Chenreddy and Erick Delage. End-to-end conditional robust optimization. arXiv preprint
> > arXiv:2403.04670, 2024.

---

> > > ### Author Response · Authors · 2025-10-17
> > > **Response to Reviewer A4ot, Part 1/2**
> > >
> > > We thank the reviewer for suggesting the comparison against the Chenreddy and Delage (2024) paper. However, we have some reservations about implementing such a comparison. We wrote the following in a footnote in our initial submission:
> > >
> > > > As of the time of writing, the approach of Chenreddy & Delage (2024) also suffers from substantial inconsistencies between
> > > [their code implementation](https://github.com/Achenred/End-to-end-CRO/) and the equations from their paper. In particular, the conditional coverage loss proposed in their
> > > paper is not implementable, as it will (almost surely) yield zero gradients.
> > >
> > > To be more explicit, the issue is two-fold. First, the gradient of the conditional coverage loss is always either zero or undefined. Second, to "overcome" this issue in their code, they implement a different loss function that does not actually account for conditional coverage. We explain these two issues in more depth below.
> > >
> > >
> > > ### Issues with the Chenreddy method
> > >
> > > First, the gradient of the conditional coverage loss $\nabla\_\theta \hat{\mathcal{L}}\_{CC}$ as written in Section 5.2 of their paper is always either zero or undefined. Specifically, the gradient involves the factor $\nabla\_\theta y^j$ where $y^j := \mathbf{1}\{\xi^j \in \mathcal{U}\_\theta(\psi^j)\}$. Here, $\psi^j$ is the input to their model, $\mathcal{U}\_\theta(\psi^j)$ is the predicted uncertainty set, and $\xi^j$ is the target.
> > >
> > > Clearly, the gradient $\nabla_\theta y^j$ is 0 as long as $\xi^j$ is not on the boundary of the set $\mathcal{U}_\theta(\psi^j)$. In the case that $\xi^j$ is on the boundary, then the gradient may be either zero or undefined. Thus, their Task based Set (TbS) approach, which does not use a conditional coverage loss, and the Dual Task based Set (DTbS) approach, which uses the conditional coverage loss, are actually equivalent, if we follow their equations exactly.
> > >
> > > However, the results in their paper show a significant improvement in coverage with DTbS compared to TbS. We reached out to author Chenreddy to seek clarification on this issue, and he responded by describing how their code implementation differs from the equations in the paper. Specifically, they implemented a straight-through estimator (STE):
> > >
> > > > Specifically, we use:
> > > >
> > > > yi+σ(s(ξi,ψi,U))−detach(σ(s(ξi,ψi,U))
> > > >
> > > > where detach(.) prevents gradients from flowing through the second sigmoid term. The function σ(s(ξi,ψi,U)) applies the sigmoid to the distance of the observation from the set U. . Since detach(.) removes the gradient in the second term, the forward computation remains unchanged. However, in the backward pass, σ(s) contributes a gradient, effectively making, ∇θL≈∇θσ(s).
> > > >
> > > > _From email correspondence with Abhilash Chenreddy_
> > >
> > > A look at their code (see either [this line](https://github.com/Achenred/End-to-end-CRO/blob/99820cd368baeae468248330b423456740194b2d/EllipsoidalConformalMTR-main/EllipsoidalConformalMTR-main/code/utils.py#L541) or [another line](https://github.com/Achenred/End-to-end-CRO/blob/99820cd368baeae468248330b423456740194b2d/models/utils.py#L774), as it's unclear which file is relevant) suggests that what was implemented is actually
> > >
> > > $$
> > >     \frac{1}{M} \sum_{i=1}^M y^i - \text{detach}(1-\sigma(s^i)) + (1-\sigma(s^i))
> > > $$
> > >
> > > where $s^i$ is the Mahalanobis distance between the label $\xi^i$ and the center of predicted ellipsoid $\mathcal{U}_\theta(\psi^i)$. This expression seems to be meant to approximate the coverage of the uncertainty set. For the gradient w.r.t. $\theta$, the first term $y^i$ has zero (or undefined) gradient, as explained above. The second term is detached, so it contributes zero gradient. Thus, the only gradient comes from the final term $1 - \sigma(s^i)$. We make two remarks about this term:
> > >
> > > 1. The term $1 - \sigma(s^i)$ monotonically increases as the Mahalanobis distance $s^i$ decreases. Thus, decreasing the distance will increase the value of this "coverage" expression.
> > > 2. However, the term $1-\sigma(s^i)$ does not appear to depend on the parametric binary classifier $g_\phi$ meant to approximate $\mathbb{P}(\xi^i \in \mathcal{U}_\theta(\psi^i))$. Thus, their code implementation is incongruent with Section 5.2 in the Chenreddy & Delage paper, which proposes to approximate the conditional coverage loss with a parametric model.
> > >
> > > In other words, their code implementation does not appear to follow the conditional coverage loss, but is instead equivalent to the heuristic of minimizing the distance between the predicted uncertainty set and the target. This is substantively different from the "regression-based conditional coverage loss" described in Section 5.2 of their paper.

---

> > > ### Author Response · Authors · 2025-10-17
> > > **Response to Reviewer A4ot, Part 2/2**
> > >
> > > ### Possible comparisons
> > >
> > > In light of the discussion above, we find it difficult to implement a comparison against the Chenreddy & Delage paper.
> > >
> > > If we implement the exact equations from their paper (leading to zero gradient from the conditional coverage loss), then we would be effectively implementing their TbS baseline, which they already show achieves very poor coverage.
> > >
> > > If we were instead to try to follow their code, then the comparison would be against the heuristic of adding a loss term that minimizes the distance between the predicted uncertainty set and the target, which is substantively different from the approach described in their paper.
> > >
> > > Finally, when we tried running their GitHub code as-is, we found that it generates numerous errors.
> > >
> > > While we would be willing to implement a comparison against one of these two methods if the reviewer deems it worthwhile, we feel that any such comparison would need to be presented with the caveats that our comparison is either not against the method proposed in the *paper* by Chenreddy & Delage, or is not against the method implemented in *code* by Chenreddy & Delage, given that these two methods appear to be distinct.

---

### Decision · Action_Editor_Dr4J · 2025-10-20

**Recommendation:** Accept as is

**Additional Comments:**

Since the paper is overall high quality, clearly written and contains sufficient evidence to support claims, I recommend the paper to be accepted as is.

**Audience:**

Yes

**Audience Explanation:**

There are two aspects of this work that are likely to garner attention from members of the community. First, the integration of downstream task into the model training end-to-end is likely to be useful in applications. Second, the usage of PICNNs to form flexible uncertainty sets is a solid contribution that the community will likely find to be of interest.

**Claims And Evidence:**

Yes

**Claims Explanation:**

This paper proposes a method to incorporate decision-awareness into conformal prediction. Parameters of the model are trained end-to-end to optimize a robust task loss. The contributions of the paper include the decision-aware end-to-end training framework, the use of partially input-convex neural networks (PICNNs) to produce the uncertainty sets, and an exact gradient computation procedure. Evidence to support these claims is provided through theoretical guarantees of the validity of the procedure as well as experiments demonstrating good task loss and coverage performance on price optimization and portfolio optimization problems. The reviewers were in agreement that the evidence was sufficient to support the claims of the paper.